# Neural Frailty Machine: Beyond proportional hazard assumption in neural survival regressions

Ruofan Wu[*][†], Jiawei Qiao[*][‡], Mingzhe Wu[§],
Wen Yu[‡], Ming Zheng[‡], Tengfei Liu[†], Tianyi Zhang[†], and Weiqiang Wang[†]

[†]Ant Group
[‡]Fudan University
[§]Coupang
{ruofan.wrf, aaron.ltf, zty113091, weiqiang.wwq}@antgroup.com
jeremyqjw@163.com, wumingzhe.darcy@gmail.com, {wenyu, mingzheng}@fudan.edu.cn

## Abstract

We present neural frailty machine (NFM), a powerful and flexible neural modeling framework for survival regressions. The NFM framework utilizes the classical idea of multiplicative frailty in survival analysis as a principled way of extending the proportional hazard assumption, at the same time being able to leverage the strong approximation power of neural architectures for handling nonlinear covariate dependence. Two concrete models are derived under the framework that extends neural proportional hazard models and nonparametric hazard regression models. Both models allow efficient training under the likelihood objective. Theoretically, for both proposed models, we establish statistical guarantees of neural function approximation with respect to nonparametric components via characterizing their rate of convergence. Empirically, we provide synthetic experiments that verify our theoretical statements. We also conduct experimental evaluations over 6 benchmark datasets of different scales, showing that the proposed NFM models achieve predictive performance comparable to or sometimes surpassing state-of-the-art survival models. Our code is publicly availabel at https://github.com/Rorschach1989/nfm

## 1 Introduction

Regression analysis of time-to-event data [40] has been among the most important modeling tools for clinical studies and has witnessed a growing interest in areas like corporate finance [22], recommendation systems [38], and computational advertising [71]. The key feature that differentiates time-to-event data from other types of data is that they are often *incompletely observed*, with the most prevailing form of incompleteness being the *right censoring* mechanism [40]. In the right censoring mechanism, the duration time of a sampled subject is (sometimes) only known to be larger than the observation time instead of being recorded precisely. It is well known in the community of survival analysis that even in the case of linear regression, naively discarding the censored observations produces estimation results that are statistically biased [7], at the same time losses sample efficiency if the censoring proportion is high.

Cox's proportional hazard (CoxPH) model [14] using the convex objective of negative partial likelihood [15] is the *de facto* choice in modeling right censored time-to-event data (hereafter abbreviated as censored data without misunderstandings). The model is *semiparametric* [4] in the sense that the baseline hazard function needs no parametric assumptions. The original formulation of CoxPH model

---

[*]Equal contribution

37th Conference on Neural Information Processing Systems (NeurIPS 2023).

assumes a linear form and therefore has limited flexibility since the truth is not necessarily linear. Subsequent studies extended CoxPH model to nonlinear variants using ideas from nonparametric regression [36, 8, 9], ensemble learning [37], and neural networks [25, 42]. While such extensions allowed a more flexible nonlinear dependence structure with the covariates, the learning objectives were still derived under the proportional hazards (PH) assumption, which was shown to be inadequate in many real-world scenarios [31]. The most notable case was the failure of modeling the phenomenon of crossing hazards [61]. It is thus of significant interest to explore extensions of CoxPH that both allow nonlinear dependence over covariates and relaxations of the PH assumption.

Frailty models [69, 21] are among the most important research topics in modern survival analysis, in that they provide a principled way of extending CoxPH model via incorporating a multiplicative random effect to capture unobserved heterogeneity. The resulting parameterization contains many useful variants of CoxPH like the proportional odds model [3], under specific choices of frailty families. While the theory of frailty models has been well-established [48, 49, 53, 45], most of them focused on the linear case. Recent developments on applying neural approaches to survival analysis [42, 46, 64, 56] have shown promising results in terms of empirical predictive performance, with most of them lacking theoretical discussions. Therefore, it is of significant interest to build more powerful frailty models via adopting techniques in modern deep learning [29] with provable statistical guarantees.

In this paper, we present a general framework for neural extensions of frailty models called the **neural frailty machine (NFM)**. Two concrete neural architectures are derived under the framework: The first one adopts the proportional frailty assumption, allowing an intuitive interpretation of the neural CoxPH model with a multiplicative random effect. The second one further relaxes the proportional frailty assumption and could be viewed as an extension of nonparametric hazard regression (NHR) [13, 44], sometimes referred to as "fully neural" models under the context of neural survival analysis [52]. We summarize our contributions as follows.

- We propose the neural frailty machine (NFM) framework as a principled way of incorporating unobserved heterogeneity into neural survival regression models. The framework includes many commonly used survival regression models as special cases.

- We derive two model architectures based on the NFM framework that extend neural CoxPH models and neural NHR models. Both models allow stochastic training and scale to large datasets.

- Theoretically, we show *statistical correctness* of the two proposed models via characterizing the rates of convergence of the proposed nonparametric function estimators. The proof technique is different from previous theoretical studies on neural survival analysis and is applicable to many other types of neural survival models.

- Empirically, we verify the *empirical efficacy* of the proposed framework via conducting extensive studies on various benchmark datasets at different scales. Under standard performance metrics, both models are empirically shown to perform competitively, matching or sometimes outperforming state-of-the-art neural survival models.

## 2 Related works

### 2.1 Nonlinear extensions of CoxPH

Most nonlinear extensions of CoxPH model stem from the equivalence of partial likelihood and semiparametric profile likelihood [50] of CoxPH model, resulting in nonlinear variants that essentially replaces the linear term in partial likelihood with nonlinear variants: [36] used smoothing splines, [8, 9] used local polynomial regression [24]. The empirical success of tree-based models inspired subsequent developments like [37] that equip tree-based models such as gradient boosting trees and random forests with losses in the form of negative log partial likelihood. Early developments of neural survival analysis [25] adopted similar extension strategies and obtained neural versions of partial likelihood. Later attempts [42] suggested using the successful practice of stochastic training which is believed to be at the heart of the empirical success of modern neural methods [34]. However, stochastic training under the partial likelihood objective is highly non-trivial, as mini-batch versions of log partial likelihood [42] are no longer valid stochastic gradients of the full-sample log partial likelihood [64].

## 2.2 Beyond CoxPH in survival analysis

In linear survival modeling, there are standard alternatives to CoxPH such as the accelerated failure time (AFT) model [7, 73], the extended hazard regression model [23], and the family of linear transformation models [74]. While these models allow certain types of nonlinear extensions, the resulting form of (conditional) hazard function is still restricted to be of a specific form. The idea of nonparametric hazard regression (NHR) [13, 44, 63] further improves the flexibility of nonparametric survival analysis via directly modeling the conditional hazard function by nonparametric regression techniques such as spline approximation. Neural versions of NHR have been developed lately such as the CoxTime model [46]. [56] used a neural network to approximate the conditional survival function and could be thus viewed as another trivial extension of NHR.

Aside from developments in NHR, [47] proposed a discrete-time model with its objective being a mix of the discrete likelihood and a rank-based score; [75] proposed a neural version of the extended hazard model, unifying both neural CoxPH and neural AFT model; [64] used an ODE approach to model the hazard and cumulative hazard functions.

## 2.3 Theoretical justification of neural survival models

Despite the abundance of neural survival models, assessment of their theoretical properties remains nascent. In [76], the authors developed minimax theories of partially linear cox model using neural networks as the functional approximator. [75] provided convergence guarantees of neural estimates under the extended hazard model. The theoretical developments therein rely on specific forms of objective function (partial likelihood and kernel pseudo-likelihood) and are not directly applicable to the standard likelihood-based objective which is frequently used in survival analysis.

# 3 Methodology

## 3.1 The neural frailty machine framework

Let $\tilde{T} \geq 0$ be the interested event time with survival function denoted by $S(t) = \mathbb{P}(\tilde{T} > t)$ associated with a feature(covariate) vector $Z \in \mathbb{R}^d$. Suppose that $\tilde{T}$ is a continuous random variable and let $f(t)$ be its density function. Then $\lambda(t) = f(t)/S(t)$ is the hazard function and $\Lambda(t) = \int_0^t \lambda(s)ds$ is the cumulative hazard function. Aside from the covariate $Z$, we use a positive scalar random variable $\omega \in \mathbb{R}^+$ to express the unobserved heterogeneity corresponding to individuals, or *frailty*. [2]. In this paper we will assume the following generating scheme of $\tilde{T}$ via specifying its conditional hazard function:

$$\lambda(t|Z, \omega) = \omega\widetilde{\nu}(t, Z). \tag{1}$$

Here $\widetilde{\nu}$ is an unspecified non-negative function, and we let the distribution of $\omega$ be parameterized by a one-dimensional parameter $\theta \in \mathbb{R}$. [3] The formulation (1) is quite general and contains several important models in both traditional and neural survival analysis:

1. When $\omega$ follows parametric distributional assumptions, and $\widetilde{\nu}(t, Z) = \lambda(t)e^{\beta^\top Z}$, (1) reduces to the standard proportional frailty model [45]. A special case is when $\omega$ is degenerate, i.e., it has no randomness, then the model corresponds to the classic CoxPH model.

2. When $\omega$ is degenerate and $\widetilde{\nu}$ is arbitrary, the model becomes equivalent to nonparametric hazard regression (NHR) [13, 44]. In NHR, the function parameter of interest is usually the logarithm of the (conditional) hazard function.

In this paper we construct neural approximations to the logarithm of $\widetilde{\nu}$, i.e., $\nu(t, Z) = \log \widetilde{\nu}(t, Z)$. The resulting models are called **Neural Frailty Machines (NFM)**. Depending on the prior knowledge of the function $\nu$, we propose two function approximation schemes:

---

[2]For example in medical biology, it was observed that genetically identical animals kept in as similar an environment as possible will typically not behave the same upon exposure to environmental carcinogens [6]

[3]The choice of one-dimensional frailty family is mostly for simplicity and clearness of theoretical derivations. Note that there exist multi-dimensional frailty families like the PVF family [69]. Generalizing our theoretical results to such kinds of families would require additional sets of regularity conditions, and will be left to future explorations.

**The proportional frailty (PF) scheme** assumes the dependence of $\nu$ on event time and covariates to be completely *decoupled*, i.e.,

$$\nu(t, Z) = h(t) + m(Z). \tag{2}$$

Proportional-style assumption over hazard functions has been shown to be a useful inductive bias in survival analysis. We will treat both $h$ and $m$ in (2) as function parameters, and device two multi-layer perceptrons (MLP) to approximate them separately.

**The fully neural (FN) scheme** imposes no a priori assumptions over $\nu$ and is the most general version of NFM. It is straightforward to see that the most commonly used survival models, such as CoxPH, AFT[73], EH[75], or PF models are included in the proposed model space as special cases. We treat $\nu = \nu(t, Z)$ as the function parameter with input dimension $d + 1$ and use a multi-layer perceptron (MLP) as the function approximator to $\nu$. Similar approximation schemes with respect to the hazard function have been proposed in some recent works [52, 56], referred to as "fully neural approaches" without theoretical characterizations.

**The choice of frailty family** There are many commonly used families of frailty distributions [45, 21, 69], among which the most popular one is the *gamma frailty*, where $\omega$ follows a gamma distribution with mean 1 and variance $\theta$. We briefly introduce some other types of frailty families in appendix A.

## 3.2 Parameter learning under censored observations

In time-to-event modeling scenarios, the event times are typically observed under right censoring. Let $C$ be the right censoring time which is assumed to be conditionally independent of the event time $\tilde{T}$ given $Z$, i.e., $\tilde{T} \perp\!\!\!\perp C | Z$. In data collection, one can observe the minimum of the survival time and the censoring time, that is, observe $T = \tilde{T} \wedge C$ as well as the censoring indicator $\delta = I(\tilde{T} \leqslant C)$, where $a \wedge b = \min(a, b)$ for constants $a$ and $b$ and $I(\cdot)$ stands for the indicator function. We assume $n$ independent and identically distributed (i.i.d.) copies of $(T, \delta, Z)$ are used as the training sample $(T_i, \delta_i, Z_i), i \in [n]$, where we use $[n]$ to denote the set $\{1, 2, \ldots, n\}$. Additionally, we assume the unobserved frailties are independent and identically distributed, i.e., $\omega_i \overset{\text{i.i.d.}}{\sim} f_\theta(\omega), i \in [n]$. Next, we derive the learning procedure based on the **observed log-likelihood (OLL)** objective under both PF and FN scheme. To obtain the observed likelihood, we first integrate the conditional survival function given the frailty:

$$S(t|Z) = \mathbb{E}_{\omega \sim f_\theta} \left[ e^{-\omega \int_0^t e^{\nu(s,Z)} ds} \right] =: e^{-G_\theta \left( \int_0^t e^{\nu(s,Z)} ds \right)}. \tag{3}$$

Here the *frailty transform* $G_\theta(x) = -\log \left( \mathbb{E}_{\omega \sim f_\theta} [e^{-\omega x}] \right)$ is defined as the negative of the logarithm of the Laplace transform of the frailty distribution. The conditional cumulative hazard function is thus $\Lambda(t|Z) = G_\theta(\int_0^t e^{\nu(s,Z)} ds)$. For the PF scheme of NFM, we use two MLPs $\widehat{h} = \widehat{h}(t; \mathbf{W}^h, \mathbf{b}^h)$ and $\widehat{m} = \widehat{m}(Z; \mathbf{W}^m, \mathbf{b}^h)$ as function approximators to $\nu$ and $m$, parameterized by $(\mathbf{W}^h, \mathbf{b}^h)$ and $(\mathbf{W}^m, \mathbf{b}^m)$, respectively. [4] According to standard results on censored data likelihood [40], we write the learning objective under the PF scheme as:

$$\mathcal{L}(\mathbf{W}^h, \mathbf{b}^h, \mathbf{W}^m, \mathbf{b}^m, \theta)$$
$$= \frac{1}{n} \left[ \sum_{i \in [n]} \delta_i \log g_\theta \left( e^{\widehat{m}(Z_i)} \int_0^{T_i} e^{\widehat{h}(s)} ds \right) + \delta_i \widehat{h}(T_i) + \delta_i \widehat{m}(Z_i) - G_\theta \left( e^{\widehat{m}(Z_i)} \int_0^{T_i} e^{\widehat{h}(s)} ds \right) \right]. \tag{4}$$

Here we define $g_\theta(x) = \frac{\partial}{\partial x} G_\theta(x)$. Let $(\widehat{\mathbf{W}}_n^h, \widehat{\mathbf{b}}_n^h, \widehat{\mathbf{W}}_n^m, \widehat{\mathbf{b}}_n^m, \widehat{\theta}_n)$ be the maximizer of (4) and further denote $\widehat{h}_n(t) = \widehat{h}(t; \widehat{\mathbf{W}}_n^h, \widehat{\mathbf{b}}_n^h)$ and $\widehat{m}_n(Z) = \widehat{m}(Z; \widehat{\mathbf{W}}_n^m, \widehat{\mathbf{b}}_n^m)$. The resulting estimators for conditional cumulative hazard and survival functions are:

$$\widehat{\Lambda}_{\mathsf{PF}}(t|Z) = G_{\widehat{\theta}_n} \left( \int_0^t e^{\widehat{h}_n(s) + \widehat{m}_n(Z)} ds \right), \qquad \widehat{S}_{\mathsf{PF}}(t|Z) = e^{-\widehat{\Lambda}_{\mathsf{PF}}(t|Z)}, \tag{5}$$

---

[4]Here we adopt the conventional notation that $\mathbf{W}$ is the collection of the weight matrices of the MLP in all layers, and $\mathbf{b}$ corresponds to the collection of the bias vectors in all layers.

For the FN scheme, we use $\widehat{\nu} = \widehat{\nu}(t, Z; \mathbf{W}^\nu, \mathbf{b}^\nu)$ to approximate $\nu(t, Z)$ parameterized by $(\mathbf{W}^\nu, \mathbf{b}^\nu)$. The OLL objective is written as:

$$
\begin{aligned}
&\mathcal{L}(\mathbf{W}^\nu, \mathbf{b}^\nu, \theta) \\
&= \frac{1}{n} \left[ \sum_{i \in [n]} \delta_i \log g_\theta \left( \int_0^{T_i} e^{\widehat{\nu}(s, Z_i; \mathbf{W}^\nu, \mathbf{b}^\nu)} ds \right) + \delta_i \widehat{\nu}(T_i, Z_i; \mathbf{W}^\nu, \mathbf{b}^\nu) - G_\theta \left( \int_0^{T_i} e^{\widehat{\nu}(s, Z_i; \mathbf{W}^\nu, \mathbf{b}^\nu)} ds \right) \right].
\end{aligned}
\tag{6}
$$

Let $(\widehat{\mathbf{W}}_n^\nu, \widehat{\mathbf{b}}_n^\nu, \widehat{\theta}_n)$ be the maximizer of (6), and further denote $\widehat{\nu}_n(t, Z) = \widehat{\nu}(t, Z; \widehat{\mathbf{W}}_n^\nu, \widehat{\mathbf{b}}_n^\nu)$. The conditional cumulative hazard and survival functions are therefore estimated as:

$$
\widehat{\Lambda}_{\mathsf{FN}}(t|Z) = G_{\widehat{\theta}_n} \left( \int_0^t e^{\widehat{\nu}_n(s, Z)} ds \right), \qquad \widehat{S}_{\mathsf{FN}}(t|Z) = e^{-\widehat{\Lambda}_{\mathsf{FN}}(t|Z)}.
\tag{7}
$$

The evaluation of objectives like (6) and its gradient requires computing a definite integral of an exponentially transformed MLP function. Instead of using exact computations that are available for only a restricted type of activation functions and network structures, we use numerical integration for such kinds of evaluations, using the method of Clenshaw-Curtis quadrature [5], which has shown competitive performance and efficiency in recent applications to monotonic neural networks [68].

*Remark* 3.1. The interpretation of frailty terms differs in the two schemes. In the PF scheme, introducing the frailty effect strictly increases the modeling capability (i.e., the capability of modeling crossing hazard) in comparison to CoxPH or neural variants of CoxPH [45]. In the FN scheme, it is arguable that in the i.i.d. case, the marginal hazard function is a reparameterization of the hazard function in the context of NHR. Therefore, we view the incorporation of frailty effect as injecting a domain-specific inductive bias that has proven to be useful in survival analysis and time-to-event regression modeling and verify this claim empirically in section 5.2. Moreover, frailty becomes especially helpful when handling correlated or clustered data where the frailty term is assumed to be shared among certain groups of individuals [53]. Extending NFM to such scenarios is valuable and we left it to future explorations.

## 4 Statistical guarantees

In this section, we present statistical guarantees regarding both NFM estimates in the sense of nonparametric regression [65], where we obtain rates of convergence to the ground truth function parameters (which is frequently referred to as the *true parameter* in statistics literature). The results in this section is interpreted as showing the *statistical correctness* of our approach.

**Proof strategy** Our proof technique is based on the method of sieves [60, 59, 11] that views neural networks as a special kind of nonlinear sieve [11] that satisfies desirable approximation properties [72]. Our strategy is different from previous theoretical works on neural survival models [75, 76] where the developments implicitly requires the loss function to be well-controlled by the $L_2$ loss and is therefore not directly applicable to our model due to the flexibility in choosing the frailty transform. Since both models produce estimates of function parameters, we need to specify a suitable function space to work with. Here we choose the following Hölder ball as was also used in previous works on nonparametric estimation using neural networks [57, 26, 76]

$$
\mathcal{W}_M^\beta(\mathcal{X}) = \left\{ f : \max_{\alpha: |\alpha| \le \beta} \operatorname*{esssup}_{x \in \mathcal{X}} |D^\alpha(f(x))| \le M \right\},
\tag{8}
$$

where the domain $\mathcal{X}$ is assumed to be a subset of $d$-dimensional euclidean space. $\alpha = (\alpha_1, \dots, \alpha_d)$ is a $d$-dimensional tuple of nonnegative integers satisfying $|\alpha| = \alpha_1 + \cdots + \alpha_d$ and $D^\alpha f = \frac{\partial^{|\alpha|} f}{\partial x_1^{\alpha_1} \cdots x_d^{\alpha_d}}$ is the weak derivative of $f$. Now assume that $M$ is a reasonably large constant, and let $\Theta$ be a closed interval over the real line. We make the following assumptions for the *true parameters* under both schemes:

**Condition 4.1** (True parameter, PF scheme). *The euclidean parameter $\theta_0 \in \Theta \subset \mathbb{R}$, and the two function parameters $m_0 \in \mathcal{W}_M^\beta([-1, 1]^d), h_0 \in \mathcal{W}_M^\beta([0, \tau])$, and $\tau > 0$ is the ending time of the study duration, which is usually adopted in the theoretical studies in survival analysis [67].*

**Condition 4.2** (True parameter, FN scheme). *The euclidean parameter $\theta_0 \in \Theta \subset \mathbb{R}$, and the function parameter $\nu_0 \in \mathcal{W}_M^\beta([0, \tau] \times [-1, 1]^d)$,*

Next, we construct sieve spaces for function parameter approximation via restricting the complexity of the MLPs to "scale" with the sample size $n$.

**Condition 4.3** (Sieve space, PF scheme). *The sieve space $\mathcal{H}_n$ is constructed as a set of MLPs satisfying $\widehat{h} \in \mathcal{W}^{\beta}_{M_h}([0, \tau])$, with depth of order $O(\log n)$ and total number of parameters of order $O(n^{\frac{1}{\beta+d}} \log n)$. The sieve space $\mathcal{M}_n$ is constructed as a set of MLPs satisfying $\widehat{m} \in \mathcal{W}^{\beta}_{M_m}([-1, 1]^d)$, with depth of order $O(\log n)$ and total number of parameters of order $O(n^{\frac{d}{\beta+d}} \log n)$. Here $M_h$ and $M_m$ are sufficiently large constants such that every function in $\mathcal{W}^{\beta}_M([-1, 1]^d)$ and $\mathcal{W}^{\beta}_M([0, \tau])$ could be accurately approximated by functions inside $\mathcal{H}_n$ and $\mathcal{M}_n$, according to [72, Theorem 1].*

**Condition 4.4** (Sieve space, FN scheme). *The sieve space $\mathcal{V}_n$ is constructed as a set of MLPs satisfying $\widehat{\nu} \in \mathcal{W}^{\beta}_{M_\nu}([0, \tau])$, with depth of order $O(\log n)$ and total number of parameters of order $O(n^{\frac{d+1}{\beta+d+1}} \log n)$. Here $M_\nu$ is a sufficiently large constant such that $\mathcal{V}_n$ satisfies approximation properties, analogous to condition 4.3.*

For technical reasons, we will assume the nonparametric function estimators are constrained to fall inside the corresponding sieve spaces, i.e., $\widehat{h}_n \in \mathcal{H}_n$, $\widehat{m}_n \in \mathcal{M}_n$ and $\widehat{\nu} \in \mathcal{V}_n$. This will not affect the implementation of optimization routines as was discussed in [26]. Furthermore, we restrict the estimate $\widehat{\theta}_n \in \Theta$ in both PF and FN schemes.

Additionally, we need the following regularity condition on the function $G_\theta(x)$:

**Condition 4.5.** $G_\theta(x)$ *is viewed as a bivariate function $G : \Theta \times \mathcal{B} \mapsto \mathbb{R}$, where $\mathcal{B}$ is a compact set on $\mathbb{R}$. The functions $G_\theta(x), \frac{\partial}{\partial \theta} G_\theta(x), \frac{\partial}{\partial x} G_\theta(x), \log g_\theta(x), \frac{\partial}{\partial \theta} \log g_\theta(x), \frac{\partial}{\partial x} \log g_\theta(x)$ are bounded on $\Theta \times \mathcal{B}$.*

We define two metrics that measures convergence of parameter estimates: For the PF scheme, let $\phi_0 = (h_0, m_0, \theta_0)$ be the true parameters and $\widehat{\phi}_n = (\widehat{h}_n, \widehat{m}_n, \widehat{\theta}_n)$ be the estimates. We abbreviate $\mathbb{P}_{\phi_0, Z=z}$ as the conditional probability distribution of $(T, \delta)$ given $Z = z$ under the true parameter, and $\mathbb{P}_{\widehat{\phi}_n, Z=z}$ as the conditional probability distribution of $(T, \delta)$ given $Z = z$ under the estimates. Define the following metric

$$d_{\mathsf{PF}}\left(\widehat{\phi}_n, \phi_0\right) = \sqrt{\mathbb{E}_{z \sim \mathbb{P}_Z}\left[H^2(\mathbb{P}_{\widehat{\phi}_n, Z=z} \parallel \mathbb{P}_{\phi_0, Z=z})\right]}, \tag{9}$$

where $H^2(\mathbb{P} \parallel \mathbb{Q}) = \int \left(\sqrt{d\mathbb{P}} - \sqrt{d\mathbb{Q}}\right)^2$ is the squared Hellinger distance between probability distributions $\mathbb{P}$ and $\mathbb{Q}$. The case for the FN scheme is similar: Let $\psi_0 = (\nu_0, \theta_0)$ be the parameters and $\widehat{\nu}_n = (\widehat{\nu}_n, \widehat{\theta}_n)$ be the estimates. Analogous to the definitions above, we define $\mathbb{P}_{\psi_0, Z=z}$ as the true conditional distribution given $Z = z$, and $\mathbb{P}_{\widehat{\psi}_n, Z=z}$ be the estimated conditional distribution, we will use the following metric in the FN scheme:

$$d_{\mathsf{FN}}\left(\widehat{\psi}_n, \psi_0\right) = \sqrt{\mathbb{E}_{z \sim \mathbb{P}_Z}\left[H^2(\mathbb{P}_{\widehat{\psi}_n, Z=z} \parallel \mathbb{P}_{\psi_0, Z=z})\right]}. \tag{10}$$

Now we state our main theorems. We denote $\mathbb{P}$ as the data generating distribution and use $\widetilde{O}$ to hide poly-logarithmic factors in the big-O notation.

**Theorem 4.6** (Rate of convergence, PF scheme). *In the PF scheme, under condition 4.1, 4.3, 4.5, we have that $d_{\mathsf{PF}}\left(\widehat{\phi}_n, \phi_0\right) = \widetilde{O}_{\mathbb{P}}\left(n^{-\frac{\beta}{2\beta+2d}}\right)$.*

**Theorem 4.7** (Rate of convergence, FN scheme). *In the FN scheme, under condition 4.2, 4.4, 4.5, we have that $d_{\mathsf{FN}}\left(\widehat{\psi}_n, \psi_0\right) = \widetilde{O}_{\mathbb{P}}\left(n^{-\frac{\beta}{2\beta+2d+2}}\right)$.*

*Remark* 4.8. The idea of using Hellinger distance to measure the convergence rate of sieve MLEs was proposed in [70]. Obtaining rates under a stronger topology such as $L_2$ is possible if the likelihood function satisfies certain conditions such as the curvature condition [26]. However, such kind of conditions is in general too stringent for likelihood-based objectives, instead, we use Hellinger distance that has minimal requirements. Consequently, our proof strategy is applicable to many other survival models that rely on neural function approximation such as [56], with some modification to the regularity conditions. For proper choices of metrics in sieve theory, see also the discussion in [11, Chapter 2].

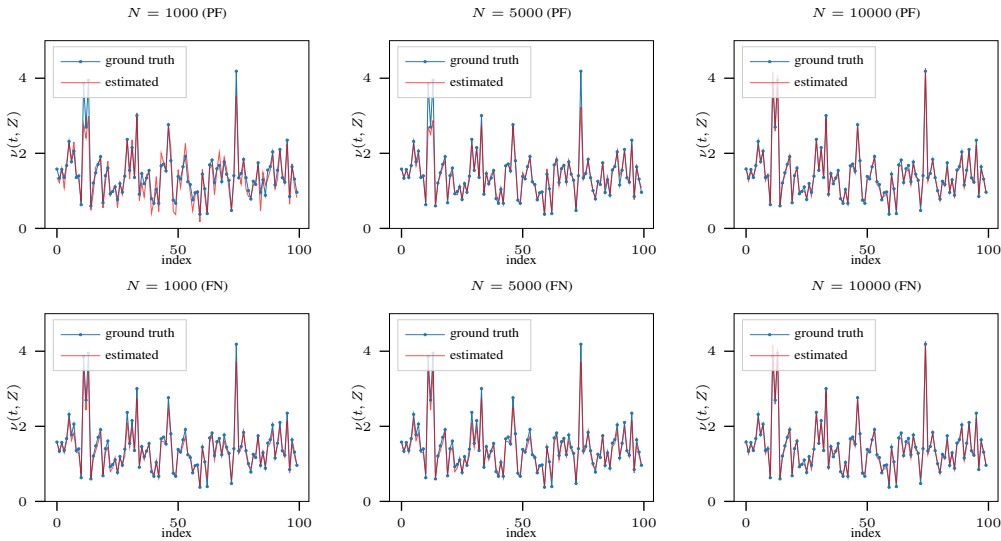

Figure 1: Visualizations of synthetic data results under the NFM framework. The plots in the first row compare the empirical estimates of the nonparametric component $\nu(t, Z)$ against its true value evaluated on 100 hold-out points, under the PF scheme. The plots in the second row are obtained using the FN scheme, with analogous semantics to the first row.

## 5 Experiments

In this section, we report the empirical performance of NFM, we will focus on the following two research questions:

**RQ1(Verfication of statistical correctness):** The results listed in section characterized the convergence results *in theory*, providing a crude guide on the number of samples required for an accurate estimate. Nonetheless, theoretical rates are often pessimistic, thus we want to investigate **whether a moderate number of sample size suffices for good approximation**.

**RQ2(Assessment of empirical efficacy):** While NFM is theoretically sound in terms of *estimation accuracy*, the theory we have developed does not necessarily guarantee its *empirical efficacy* as a method of doing prognosis. It is therefore valuable to inspect **how useful NFM is regarding real-world predictive tasks in survival analysis**.

### 5.1 Synthetic experiments

To answer RQ1, we conduct synthetic experiments to check the empirical convergence. Specifically, we investigate the empirical recovery of underlying ground truth parameters under various level of sample size.
**Ground truth** We set the true underlying model to be a nonlinear gamma-frailty model with a 5-dimensional feature. We generate three training datasets of different scales, with $n \in \{1000, 5000, 10000\}$. The assessment will be made on a fixed test sample of 100 hold-out points that are independently drawn from the generating scheme of the event time. A censoring mechanism is applied such that the censoring ratio is around $40\%$ for each dataset. The precise form of the frailty model as well as the generating distribution of the feature vectors are detailed in appendix C.2.
**Empirical recovery results** We report the empirical recovery of the nonlinear component $\nu(t, Z)$ regarding the hold-out test set in in figure 1. We observe from graphical illustrations that under a moderate sample size $n = 1000$, NFM already exhibits satisfactory recovery for a (relatively low-dimensional) feature space, which is the prevailing case in most public benchmark datasets. We also present additional assessments about: (i) The recovery of $m(Z)$ when using PF scheme in appendix D.1, (ii) The recovery of survival functions under both PF and FN scheme in appendix D.2, and (iii) The numerical recovery results of survival function in appendix D.3.

Table 1: Survival prediction results measured in IBS and INBLL metric (%) on four small-scale survival datasets. In each column, the **boldfaced** score denotes the best result and the underlined score represents the second-best result (both in mean).

| Model | METABRIC | | RotGBSG | | FLCHAIN | | SUPPORT | |
|---|---|---|---|---|---|---|---|---|
| | IBS | INBLL | IBS | INBLL | IBS | INBLL | IBS | INBLL |
| CoxPH | $16.46_{\pm0.90}$ | $49.57_{\pm2.66}$ | $18.25_{\pm0.44}$ | $53.76_{\pm1.11}$ | $10.05_{\pm0.38}$ | $33.18_{\pm1.16}$ | $20.54_{\pm0.38}$ | $59.58_{\pm0.86}$ |
| GBM | $16.61_{\pm0.82}$ | $49.87_{\pm2.44}$ | $17.83_{\pm0.44}$ | $52.78_{\pm1.11}$ | $\underline{9.98}_{\pm0.37}$ | $\underline{32.88}_{\pm1.05}$ | $19.18_{\pm0.39}$ | $56.46_{\pm0.10}$ |
| RSF | $16.62_{\pm0.64}$ | $49.61_{\pm1.54}$ | $17.89_{\pm0.42}$ | $52.77_{\pm1.01}$ | $\mathbf{9.96}_{\pm0.37}$ | $32.92_{\pm1.05}$ | $\underline{19.11}_{\pm0.40}$ | $\underline{56.28}_{\pm1.00}$ |
| DeepSurv | $16.55_{\pm0.93}$ | $49.85_{\pm3.02}$ | $17.80_{\pm0.49}$ | $52.62_{\pm1.25}$ | $10.09_{\pm0.38}$ | $33.28_{\pm1.15}$ | $19.20_{\pm0.41}$ | $56.48_{\pm1.08}$ |
| CoxTime | $16.54_{\pm0.83}$ | $49.67_{\pm2.67}$ | $17.80_{\pm0.58}$ | $52.56_{\pm1.47}$ | $10.28_{\pm0.45}$ | $34.18_{\pm1.53}$ | $19.17_{\pm0.40}$ | $56.45_{\pm1.10}$ |
| DeepHit | $17.50_{\pm0.83}$ | $52.10_{\pm2.16}$ | $19.61_{\pm0.38}$ | $56.67_{\pm1.10}$ | $11.83_{\pm0.39}$ | $37.72_{\pm1.02}$ | $20.66_{\pm0.32}$ | $60.06_{\pm0.72}$ |
| DeepEH | $16.56_{\pm0.65}$ | $49.42_{\pm1.53}$ | $17.62_{\pm0.52}$ | $\underline{52.08}_{\pm1.27}$ | $10.11_{\pm0.37}$ | $33.30_{\pm1.10}$ | $19.30_{\pm0.39}$ | $56.67_{\pm0.94}$ |
| SuMo-net | $16.49_{\pm0.83}$ | $49.74_{\pm2.21}$ | $17.77_{\pm0.47}$ | $52.62_{\pm1.11}$ | $10.07_{\pm0.40}$ | $33.20_{\pm1.10}$ | $19.40_{\pm0.38}$ | $56.87_{\pm0.96}$ |
| SODEN | $16.52_{\pm0.63}$ | $49.39_{\pm1.97}$ | $\mathbf{17.05}_{\pm0.63}$ | $\mathbf{50.45}_{\pm1.97}$ | $10.13_{\pm0.24}$ | $33.37_{\pm0.57}$ | $19.07_{\pm0.50}$ | $56.15_{\pm1.35}$ |
| SurvNode | $16.67_{\pm1.32}$ | $49.73_{\pm3.89}$ | $17.42_{\pm0.53}$ | $51.70_{\pm1.16}$ | $10.40_{\pm0.29}$ | $34.37_{\pm1.03}$ | $19.58_{\pm0.34}$ | $57.49_{\pm0.84}$ |
| DCM | $16.58_{\pm0.87}$ | $49.48_{\pm2.23}$ | $17.66_{\pm0.54}$ | $52.26_{\pm1.23}$ | $10.13_{\pm0.50}$ | $33.40_{\pm1.38}$ | $19.29_{\pm0.42}$ | $56.68_{\pm1.09}$ |
| DeSurv | $16.71_{\pm0.75}$ | $49.61_{\pm2.15}$ | $17.98_{\pm0.46}$ | $53.23_{\pm1.15}$ | $10.06_{\pm0.62}$ | $33.18_{\pm1.93}$ | $19.50_{\pm0.40}$ | $57.28_{\pm0.89}$ |
| **NFM-PF** | $\underline{16.33}_{\pm0.75}$ | $\underline{49.07}_{\pm1.96}$ | $\underline{17.60}_{\pm0.55}$ | $52.12_{\pm1.34}$ | $\mathbf{9.96}_{\pm0.39}$ | $\mathbf{32.84}_{\pm1.15}$ | $19.14_{\pm0.39}$ | $56.35_{\pm1.00}$ |
| **NFM-FN** | $\mathbf{16.11}_{\pm0.81}$ | $\mathbf{48.21}_{\pm2.04}$ | $17.66_{\pm0.52}$ | $52.41_{\pm1.22}$ | $10.05_{\pm0.39}$ | $33.11_{\pm1.10}$ | $\mathbf{18.97}_{\pm0.60}$ | $\mathbf{55.87}_{\pm1.50}$ |

## 5.2 Real-world data experiments

To answer RQ2, we conduct extensive empirical assessments over 6 benchmark datasets, comprising five survival datasets and one non-survival dataset. The survival datasets include the Molecular Taxonomy of Breast Cancer International Consortium (METABRIC) [16], the Rotterdam tumor bank and German Breast Cancer Study Group (RotGBSG)[43], the Assay Of Serum Free Light Chain (FLCHAIN) [20], the Study to Understand Prognoses Preferences Outcomes and Risks of Treatment (SUPPORT) [43], and the Medical Information Mart for Intensive Care (MIMIC-III) [39]. For all the survival datasets, the event of interest is defined as the mortality after admission. In our experiments, we view METABRIC, RotGBSG, FLCHAIN, and SUPPORT as small-scale datasets and MIMIC-III as a moderate-scale dataset. We additionally use the KKBOX dataset [46] as a large-scale evaluation. In this dataset, an event time is observed if a customer churns from the KKBOX platform. We summarize the basic statistics of all the datasets in table 3.

**Baselines** We compare NFM with 12 baselines. The first one is linear CoxPH model [14]. Gradient Boosting Machine (GBM) [27, 10] and Random Survival Forests (RSF) [37] are two tree-based nonparametric survival regression methods. DeepSurv [42] and CoxTime [46] are two models that adopt neural variants of partial likelihood as objectives. SuMo-net [56] is a neural variant of NHR. We additionally chose six latest state-of-the-art neural survival models: DeepHit [47], SurvNode [32], DeepEH [75], DCM [51], DeSurv [19] and SODEN [64]. Among the chosen baselines, DeepSurv and SuMo-net are viewed as implementations of neural CoxPH and neural NHR and are therefore of particular interest for the empirical verification of the efficacy of frailty.

**Evaluation strategy** We use two standard metrics in survival predictions for evaluating model performance: integrated Brier score (IBS) and integrated negative binomial log-likelihood (INBLL). Both metrics are derived from the following:

$$\mathcal{S}(\ell, t_1, t_2) = \int_{t_2}^{t_1} \frac{1}{n} \sum_{i=1}^{n} \left[ \frac{\ell(0, \widehat{S}(t|Z_i))I(T_i \leq t, \delta_i = 1)}{\widehat{S}_C(T_i)} + \frac{\ell(1, \widehat{S}(t|Z_i))I(T_i > t)}{\widehat{S}_C(t)} \right] dt. \quad (11)$$

Where $\widehat{S}_C(t)$ is an estimate of the survival function $S_C(t)$ of the censoring variable, obtained by the Kaplan-Meier estimate [41] of the censored observations on the test data. $\ell : \{0, 1\} \times [0, 1] \mapsto \mathbb{R}^+$ is some proper loss function for binary classification [28]. The IBS metric corresponds to $\ell$ being the square loss, and the INBLL metric corresponds to $\ell$ being the negative binomial (Bernoulli) log-likelihood [30]. Both IBS and INBLL are proper scoring rules if the censoring times and survival times are independent. [5] We additionally report the result of another widely used metric, the concordance index (C-index), in appendix D. Since all the survival datasets do not have standard

---

[5]Otherwise, one may pose a covariate-dependent model on the censoring time and use $\widehat{S}_C(t|Z)$ instead of $\widehat{S}_C(t)$. We adopt the Kaplan-Meier approach since it's still the prevailing practice in evaluations of survival predictions.

Table 2: Survival prediction results measured in IBS and INBLL metric (%) on two larger datasets. In each column, the **boldfaced** score denotes the best result and the underlined score represents the second-best result (both in mean). Two models are not reported, namely SODEN and DeepEH, as we found empirically that their computational/memory cost is significantly worse than the rest, and we fail to obtain reasonable performances over the two datasets for these two models.

| Model | MIMIC-III | | KKBOX | |
|---|---|---|---|---|
| | IBS | INBLL | IBS | INBLL |
| CoxPH | $20.40_{\pm 0.00}$ | $60.02_{\pm 0.00}$ | $12.60_{\pm 0.00}$ | $39.40_{\pm 0.00}$ |
| GBM | $17.70_{\pm 0.00}$ | $52.30_{\pm 0.00}$ | $11.81_{\pm 0.00}$ | $38.15_{\pm 0.00}$ |
| RSF | $17.79_{\pm 0.19}$ | $53.34_{\pm 0.41}$ | $14.46_{\pm 0.00}$ | $44.39_{\pm 0.00}$ |
| DeepSurv | $18.58_{\pm 0.92}$ | $55.98_{\pm 2.43}$ | $11.31_{\pm 0.05}$ | $35.28_{\pm 0.15}$ |
| CoxTime | $17.68_{\pm 1.36}$ | $52.08_{\pm 3.06}$ | $\underline{10.70}_{\pm 0.06}$ | $\underline{33.10}_{\pm 0.21}$ |
| DeepHit | $19.80_{\pm 1.31}$ | $59.03_{\pm 4.20}$ | $16.00_{\pm 0.34}$ | $48.64_{\pm 1.04}$ |
| SuMo-net | $18.62_{\pm 1.23}$ | $54.51_{\pm 2.97}$ | $11.58_{\pm 0.11}$ | $36.61_{\pm 0.28}$ |
| DCM | $18.02_{\pm 0.49}$ | $52.83_{\pm 0.94}$ | $10.71_{\pm 0.11}$ | $33.24_{\pm 0.06}$ |
| DeSurv | $18.19_{\pm 0.65}$ | $54.69_{\pm 2.83}$ | $10.77_{\pm 0.21}$ | $33.22_{\pm 0.10}$ |
| **NFM-PF** | $\mathbf{16.28}_{\pm 0.36}$ | $\mathbf{49.18}_{\pm 0.92}$ | $11.02_{\pm 0.11}$ | $35.10_{\pm 0.22}$ |
| **NFM-FN** | $\underline{17.47}_{\pm 0.45}$ | $\underline{51.48}_{\pm 1.23}$ | $\mathbf{10.63}_{\pm 0.08}$ | $\mathbf{32.81}_{\pm 0.14}$ |

train/test splits, we follow previous practice [75] that uses 5-fold cross-validation (CV): 1 fold is for testing, and 20% of the rest is held out for validation. In our experiments, we observed that a single random split into 5 folds does not produce stable results for most survival datasets. Therefore we perform 10 different CV runs for each survival dataset and report average metrics as well as their standard deviations. For the KKBOX dataset, we use the standard train/valid/test splits that are available via the `pycox` package [46] and report results based on 10 trial runs.

**Experimental setup** We follow standard preprocessing strategies [42, 46, 75] that standardize continuous features into zero mean and unit variance, and do one-hot encodings for all categorical features. We adopt MLP with ReLU activation for all function approximators, including $\widehat{h}, \widehat{m}$ in PF scheme, and $\widehat{\nu}$ in FN scheme, across all datasets, with the number of layers (depth) and the number of hidden units (width) within each layer being tunable. We tune the frailty transform over several standard choices: gamma frailty, Box-Cox transformation frailty and IGG($\alpha$) frailty, with their precise forms detailed in appendix C.3. A more detailed description of the tuning procedure, as well as training configurations for baseline models, are reported in appendix C.3.

**Results** We report experimental results of small-scale datasets in table 1, and results of two larger datasets in table 2. The proposed NFM framework achieves competitive performance which is comparable to the other state-of-the-art models. In particular, NFM attains best performance in mean on 5 of the 6 datasets, and is statistically significantly better over all the baselines at 0.05 empirical level on the MIMIC-III dataset.

**Ablation on the benefits of frailty** To better understand the additional benefits of introducing the frailty formulation, we compute the (relative) performance gain of NFM-PF and NFM-FN, against their non-frailty counterparts, namely DeepSurv [42] and SuMo-net [56]. The evaluation is conducted for all three metrics mentioned in this paper. The results are shown in table 6. The results suggest a solid improvement in incorporating frailty, as the relative increase in performance could be over 10% for both NFM models. A more detailed discussion is presented in section D.5.

# 6  Discussion and conclusion

We have introduced NFM as a flexible and powerful neural modeling framework for survival analysis, which is shown to be both statistically correct in theory, and empirically effective in predictive tasks. While our proposed framework provides a theoretically-principled tool of neural survival modeling, a few limitations and challenges need to be addressed in future works including predictive guarantees and better evaluation protocols, which we elaborate in appendix D.6.

# 7 Acknowledgements

We would like to thank professor Zhiliang Ying and professor Guanhua Fang for helpful discussions. Wen Yu's research is supported by the National Natural Science Foundation of China Grants (12071088). Ming Zheng's research is supported by the National Natural Science Foundation of China Grants (12271106).

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

## A  Examples of frailty specifications

We list several commonly used frailty models, and specify their corresponding characteristics via their frailty transform $G_\theta$:

**Gamma frailty:**  Arguably the gamma frailty is the most widely used frailty model [48, 49, 53, 69, 21], with

$$G_\theta(x) = \frac{1}{\theta}\log(1+\theta x), \theta \geq 0. \tag{12}$$

When $\theta = 0$, $G_0(x) = \lim_{\theta \to 0} G_\theta(x)$ is defined as the (pointwise) limit. A notable fact of the gamma frailty specification is that when the proportional frailty (PF) assumption (2) is met, if $\theta = 0$, the model degenerates to CoxPH. Otherwise if $\theta = 1$, the model corresponds to the proportional odds (PO) model [3].

**Box-Cox transformation frailty:**  Under this specification, we have

$$G_\theta(x) = \frac{(1+x)^\theta - 1}{\theta}, \theta \geq 0. \tag{13}$$

The case of $\theta = 0$ is defined analogously to that of gamma frailty, which corresponds to the PO model under the PF assumption. When $\theta = 1$, the model reduces to CoxPH under the PF assumption.

**IGG$(\alpha)$ frailty:**  This is an extension of gamma frailty [45] and includes other types of frailty specifications like the inverse gaussian frailty [35], with

$$G_\theta(x) = \frac{1-\alpha}{\alpha\theta}\left[\left(1 + \frac{\theta x}{1-\alpha}\right)^\alpha - 1\right], \theta \geq 0, \alpha \in [0,1). \tag{14}$$

In the one-dimensional parameter paradigm, the parameter $\alpha$ is assumed known instead of being learnable. When $\alpha = 1/2$, we obtain the gamma frailty model. When $\alpha \to 0$, the limit corresponds to the inverse Gaussian frailty.

**Satistiability of regularity condition 4.5**  In [45, Proposition 1], the authors verified the regularity condition of gamma and IGG$(\alpha)$ frailties. Using a similar argument, it is straightforward to verify the regularity of Box-Cox transformation frailty.

## B  Proofs of theorems

### B.1  Preliminary

**Additional definitions**  The theory of empirical processes [66] will be involved heavily in the proof. Therefore we briefly introduce some common notations: For a function class $\mathcal{F}$, define $N(\epsilon, \mathcal{F}, \|\cdot\|)$ to be the covering number of $\mathcal{F}$ with respect to norm $\|\cdot\|$ under radius $\epsilon$, and define $N_{[]}(\epsilon, \mathcal{F}, \|\cdot\|)$ to be the bracketing number of $\mathcal{F}$ with respect to norm $\|\cdot\|$ under radius $\epsilon$. We use VC$(\mathcal{F})$ to denote the VC-dimension of $\mathcal{F}$. Moreover, we use the notation $a \lesssim b$ to denote $a \leq Cb$ for some positive constant $C$.

Before proving theorem 4.6 and 4.7, we introduce some additional notations that will be useful throughout the proof process.

In the PF scheme, define

$$\begin{aligned}
l(T, \delta, Z; h, m, \theta) =& \delta \log g_\theta\left(e^{m(Z)}\int_0^T e^{h(s)}ds\right) + \delta h(T) + \delta m(Z) \\
&- G_\theta\left(e^{m(Z)}\int_0^T e^{h(s)}ds\right),
\end{aligned}$$

where we denote $g_\theta = G'(\theta)$. Under the definition of the sieve space stated in condition 4.3, we restate the parameter estimates as

$$\left(\widehat{h}_n, \widehat{m}_n, \widehat{\theta}_n\right) = \underset{\widehat{h}\in\mathcal{H}_n, \widehat{m}\in\mathcal{M}_n, \theta\in\Theta}{\operatorname{argmax}} \frac{1}{n}\sum_{i\in[n]} l(T_i, \delta_i, Z_i; \widehat{h}, \widehat{m}, \theta).$$

Similarly, in the FN scheme, we define

$$l(T, \delta, Z; \nu, \theta) = \delta \log g_\theta \left( \int_0^T e^{\nu(s,Z)} ds \right) + \delta \nu(T, Z) - G_\theta \left( \int_0^T e^{\nu(s,Z)} ds \right)$$

Under the definition of the sieve space stated in condition 4.4, we restate the parameter estimates as

$$\left( \widehat{\nu}_n(t, z), \widehat{\theta}_n \right) = \operatorname*{argmax}_{\widehat{\nu} \in \mathcal{V}_n, \theta \in \Theta} \frac{1}{n} \sum_{i \in [n]} l(T_i, \delta_i, Z_i; \widehat{\nu}, \theta).$$

We denote the conditional density function and survival function of the event time $\tilde{T}$ given $Z$ by $f_{\tilde{T}|Z}(t)$ and $S_{\tilde{T}|Z}(t)$, respectively. Similarly, we denote the conditional density function and survival function of the censoring time $C$ given $Z$ by $f_{C|Z}(t)$ and $S_{C|Z}(t)$. Under the assumption that $\tilde{T} \perp\!\!\!\perp C \mid Z$, the joint conditional density of the observed time $T$ and the censoring indicator $\delta$ given $Z$ can be expressed as the following:

$$
\begin{aligned}
p(T, \delta \mid Z) &= f_{\tilde{T}|Z}(T)^\delta S_{\tilde{T}|Z}(T)^{1-\delta} f_{C|Z}(T)^{1-\delta} S_{C|Z}(T)^\delta \\
&= \lambda_{\tilde{T}|Z}(T)^\delta S_{\tilde{T}|Z}(T) f_{C|Z}(T)^{1-\delta} S_{C|Z}(T)^\delta,
\end{aligned}
$$

where $\lambda_{\tilde{T}|Z}(T)$ is the conditional hazard function of the survival time $\tilde{T}$ given $Z$.

Under the model assumption of PF scheme, $p(T, \delta \mid Z)$ can be expressed by

$$p(T, \delta \mid Z; h, m, \theta) = \exp\left( l(T, \delta, Z; h, m, \theta) \right) f_{C|Z}(T)^{1-\delta} S_{C|Z}(T)^\delta.$$

For $\phi_0 = (h_0, m_0, \theta_0)$ and an estimator $\widehat{\phi} = (\widehat{h}, \widehat{m}, \widehat{\theta})$, the defined distance $d_{\mathsf{PF}}\left( \widehat{\phi}, \phi_0 \right)$ can be explicitly expresses by

$$d_{\mathsf{FN}}\left( \widehat{\psi}, \psi_0 \right) = \sqrt{\mathbb{E}_Z \left[ \int \left| \sqrt{p(T, \delta \mid Z; \widehat{h}, \widehat{m}, \widehat{\theta})} - \sqrt{p(T, \delta \mid Z; h_0, m_0, \theta_0)} \right|^2 \mu(dT \times d\delta) \right]}.$$

Here the dominating measure $\mu$ is defined such that for any (measurable) function $r(T, \delta)$

$$\int r(T, \delta) \mu(dT \times d\delta) = \int_0^\tau r(T, \delta = 1) dT + \int_0^\tau r(T, \delta = 0) dT$$

Under the model assumption of FN scheme, $p(T, \delta \mid Z)$ can be expressed by

$$p(T, \delta \mid Z; \nu, \theta) = \exp\left( l(T, \delta, Z; \nu, \theta) \right) f_{C|Z}(T)^{1-\delta} S_{C|Z}(T)^\delta.$$

For $\psi_0 = (\nu_0, \theta_0)$ and an estimator $\widehat{\psi} = (\widehat{\nu}, \widehat{\theta})$, the defined distance $d_{\mathsf{FN}}\left( \widehat{\psi}, \psi_0 \right)$ can be explicitly expresses by

$$d_{\mathsf{FN}}\left( \widehat{\psi}, \psi_0 \right) = \sqrt{\mathbb{E}_Z \left[ \int \left| \sqrt{p(T, \delta \mid Z; \widehat{\nu}, \widehat{\theta})} - \sqrt{p(T, \delta \mid Z; \nu_0, \theta_0)} \right|^2 \mu(dT \times d\delta) \right]}.$$

## B.2 Technical lemmas

The following lemmas are needed for the proof of Theorem 4.6 and 4.7. Hereafter for notational convenience, we will use $\widehat{h}, \widehat{m}$ for arbitrary elements in the corresponding sieve space listed in condition 4.3, $\widehat{\nu}$ for an arbitrary element in the sieve space listed in condition 4.4, and $\widehat{\theta}$ for an arbitrary element in $\Theta$.

**Lemma B.1.** *Under condition 4.1, 4.3, 4.5, for $(T, \delta, Z) \in [0, \tau] \times \{0, 1\} \times [-1, 1]^d$, the following terms are bounded:*

1. $l(T, \delta, Z; h_0, m_0, \theta_0)$ with true parameter $(h_0, m_0, \theta_0)$

2. $l(T, \delta, Z; \widehat{h}, \widehat{m}, \widehat{\theta})$ with parameter estimates $(\widehat{h}, \widehat{m}, \widehat{\theta})$ in any sieve space listed in condition 4.3.

**Lemma B.2.** *Under condition 4.2, 4.4, 4.5, for $(T, \delta, Z) \in [0, \tau] \times \{0, 1\} \times [-1, 1]^d$, the following terms are bounded:*

1. $l(T, \delta, Z; \nu_0, \theta_0)$ with true parameter $(\nu_0, \theta_0)$

2. $l(T, \delta, Z; \widehat{\nu}, \widehat{\theta})$ with parameter estimates $(\widehat{\nu}, \widehat{\theta})$ in any sieve space listed in condition 4.4.

**Lemma B.3.** *Under condition 4.1, 4.3, 4.5, let $(\widehat{h}, \widehat{m}, \widehat{\theta})$, $(\widehat{h}_1, \widehat{m}_1, \widehat{\theta}_1)$, and $(\widehat{h}_2, \widehat{m}_2, \widehat{\theta}_2)$ be arbitrary three parameter triples inside the sieve space defined in condition 4.3, the following two inequalities hold.*

$$\|l(T, \delta, Z; h_0, m_0, \theta_0) - l(T, \delta, Z; \widehat{h}, \widehat{m}, \widehat{\theta})\|_\infty \lesssim |\theta_0 - \widehat{\theta}| + \|h_0 - \widehat{h}\|_\infty + \|m_0 - \widehat{m}\|_\infty$$

$$\|l(T, \delta, Z; \widehat{h}_1, \widehat{m}_1, \widehat{\theta}_1) - l(T, \delta, Z; \widehat{h}_2, \widehat{m}_2, \widehat{\theta}_2)\|_\infty \lesssim |\widehat{\theta}_1 - \widehat{\theta}_2| + \|\widehat{h}_1 - \widehat{h}_2\|_\infty + \|\widehat{m}_1 - \widehat{m}_2\|_\infty.$$

**Lemma B.4.** *Under condition 4.2, 4.4, 4.5, let $(\widehat{\nu}, \widehat{\theta})$, $(\widehat{\nu}_1, \widehat{\theta}_1)$, and $(\widehat{\nu}_2, \widehat{\theta}_2)$ be arbitrary three parameter tuples inside the sieve space defined in condition 4.4,, the following inequalities hold.*

$$\|l(T, \delta, Z; \nu_0, \theta_0) - l(T, \delta, Z; \widehat{\nu}, \widehat{\theta})\|_\infty \lesssim |\theta_0 - \widehat{\theta}| + \|\nu_0 - \widehat{\nu}\|_\infty$$

$$\|l(T, \delta, Z; \widehat{\nu}_1, \widehat{\theta}_1) - l(T, \delta, Z; \widehat{\nu}_2, \widehat{\theta}_2)\|_\infty \lesssim |\widehat{\theta}_1 - \widehat{\theta}_2| + \|\widehat{\nu}_1 - \widehat{\nu}_2\|_\infty.$$

**Lemma B.5** (Approximating error of PF scheme). *In the PF scheme, for any $n$, there exists an element in the corresponding sieve space $\pi_n \phi_0 = (\pi_n h_0, \pi_n m_0, \pi_n \theta_0)$, satisfying $d_{PF}(\pi_n \phi_0, \phi_0) = O\left(n^{-\frac{\beta}{\beta + d}}\right)$.*

**Lemma B.6** (Approximating error of FN scheme). *In the FN scheme, for any $n$, there exists an element in the corresponding sieve space $\pi_n \psi = (\pi_n \nu_0, \pi_n \theta_0)$ satisfying $d_{FN}(\pi_n \psi_0, \psi_0) = O\left(n^{-\frac{\beta}{\beta + d + 1}}\right)$.*

**Lemma B.7.** *Suppose $\mathcal{F}$ is a class of functions satisfying that $N(\varepsilon, \mathcal{F}, \| \cdot \|) < \infty$ for $\forall \varepsilon > 0$. We define $\widetilde{N}(\varepsilon, \mathcal{F}, \| \cdot \|)$ to be the minimal number of $\varepsilon$-balls $B(f, \varepsilon) = \{g : \|g - f\| < \varepsilon\}$ needed to cover $\mathcal{F}$ with radius $\varepsilon$ and further constrain that $f \in \mathcal{F}$. Then we have*

$$N(\varepsilon, \mathcal{F}, \| \cdot \|) \leq \widetilde{N}(\varepsilon, \mathcal{F}, \| \cdot \|) \leq N(\frac{\varepsilon}{2}, \mathcal{F}, \| \cdot \|).$$

**Lemma B.8.** *Suppose $\mathcal{F}$ is a class of functions satisfying that $N_{[]}(\varepsilon, \mathcal{F}, \| \cdot \|_\infty) < \infty$ for $\forall \varepsilon > 0$. We define $\widetilde{N}_{[]}(\varepsilon, \mathcal{F}, \| \cdot \|_\infty)$ to be the minimal number of brackets $[l, u]$ needed to cover $\mathcal{F}$ with $\|l - u\|_\infty < \varepsilon$ and further constrain that $f \in \mathcal{F}$, $l = f - \frac{\varepsilon}{2}$ and $u = f + \frac{\varepsilon}{2}$. Then we have*

$$N_{[]}(\varepsilon, \mathcal{F}, \| \cdot \|_\infty) \leq \widetilde{N}_{[]}(\varepsilon, \mathcal{F}, \| \cdot \|_\infty) \leq N_{[]}(\frac{\varepsilon}{2}, \mathcal{F}, \| \cdot \|_\infty)$$

*Furthermore, we have $\widetilde{N}_{[]}(\varepsilon, \mathcal{F}, \| \cdot \|_\infty) = \widetilde{N}(\frac{\varepsilon}{2}, \mathcal{F}, \| \cdot \|_\infty)$.*

**Lemma B.9** (Model capacity of PF scheme). *Let $\mathcal{F}_n = \{l(T, \delta, Z; \widehat{h}, \widehat{m}, \widehat{\theta}) : \widehat{h} \in \mathcal{H}_n, \widehat{m} \in \mathcal{M}_n, \widehat{\theta} \in \Theta\}$. Under condition 4.5, with $s_h = \frac{2\beta}{2\beta + 1}$ and $s_m = \frac{2\beta}{2\beta + d}$, there exist constants $c_h$ and $c_m > 0$ such that*

$$N_{[]}(\varepsilon, \mathcal{F}_n, \| \cdot \|_\infty) \lesssim \frac{1}{\varepsilon} N(c_h \varepsilon^{1/s_h}, \mathcal{H}_n, \| \cdot \|_2) \times N(c_m \varepsilon^{1/s_m}, \mathcal{M}_n, \| \cdot \|_2).$$

**Lemma B.10** (Model capacity of FN scheme). *Let $\mathcal{G}_n = \{l(T, \delta, Z; \widehat{\nu}, \widehat{\theta}) : \widehat{\nu} \in \mathcal{V}_n, \widehat{\theta} \in \Theta\}$. Under condition 4.5, with $s_\nu = \frac{2\beta}{2\beta + d + 1}$, there exists a constant $c_\nu > 0$ such that*

$$N_{[]}(\varepsilon, \mathcal{G}_n, \| \cdot \|_\infty) \lesssim \frac{1}{\varepsilon} N(c_\nu \varepsilon^{1/s_\nu}, \mathcal{V}_n, \| \cdot \|_2).$$

## B.3 Proofs of theorem 4.6 and 4.7

*Proof of theorem 4.6.* The proof is divided into four steps.

**Step** 1   We denote $\phi_0 = (h_0, m_0, \theta_0)$ and $\widehat{\phi} = (\widehat{h}, \widehat{m}, \widehat{\theta})$, where $\widehat{h} \in \mathcal{H}_n, \widehat{m} \in \mathcal{M}_n$ and $\widehat{\theta} \in \Theta$. For arbitrary small $\varepsilon > 0$, we have that

$$\inf_{d_{\mathsf{PF}}(\widehat{\phi}, \phi_0) \geq \varepsilon} \mathbb{E}\left[l(T, \delta, Z; h_0, m_0, \theta_0) - l(T, \delta, Z; \widehat{h}, \widehat{m}, \widehat{\theta})\right]$$

$$= \inf_{d_{\mathsf{PF}}(\widehat{\phi}, \phi_0) \geq \varepsilon} \mathbb{E}_Z\left[\mathbb{E}_{T, \delta | Z}\left[\log p(T, \delta \mid Z; h_0, m_0, \theta_0) - \log p(T, \delta \mid Z; \widehat{h}, \widehat{m}, \widehat{\theta})\right]\right]$$

$$= \inf_{d_{\mathsf{PF}}(\widehat{\phi}, \phi_0) \geq \varepsilon} \mathbb{E}_Z\left[D_{\mathrm{KL}}\left(\mathbb{P}_{\widehat{\phi}, Z} \parallel \mathbb{P}_{\phi_0, Z}\right)\right]$$

Using the fact that $D_{\mathrm{KL}}\left(\mathbb{P}_{\widehat{\phi}, Z} \parallel \mathbb{P}_{\phi_0, Z}\right) \geq 2H^2(\mathbb{P}_{\widehat{\phi}, Z} \parallel \mathbb{P}_{\phi_0, Z})$. Thus, we further obtain that

$$\inf_{d_{\mathsf{PF}}(\widehat{\phi}, \phi_0) \geq \varepsilon} \mathbb{E}\left[l(T, \delta, Z; h_0, m_0, \theta_0) - l(T, \delta, Z; \widehat{h}, \widehat{m}, \widehat{\theta})\right]$$

$$\geq \inf_{d_{\mathsf{PF}}(\widehat{\phi}, \phi_0) \geq \varepsilon} 2\mathbb{E}_Z\left[H^2(\mathbb{P}_{\widehat{\phi}, Z} \parallel \mathbb{P}_{\phi_0, Z})\right]$$

$$= 2 \inf_{d_{\mathsf{PF}}(\widehat{\phi}, \phi_0) \geq \varepsilon} d_{\mathsf{PF}}^2\left(\widehat{\phi}, \phi_0\right)$$

$$\geq 2\varepsilon^2.$$

**Step** 2   Consider the following derivation.

$$\sup_{d_{\mathsf{PF}}(\widehat{\phi}, \phi_0) \leq \varepsilon} \mathrm{Var}\left[l(T, \delta, Z; h_0, m_0, \theta_0) - l(T, \delta, Z; \widehat{h}, \widehat{m}, \widehat{\theta})\right]$$

$$\leq \sup_{d_{\mathsf{PF}}(\widehat{\phi}, \phi_0) \leq \varepsilon} \mathbb{E}\left[\left(l(T, \delta, Z; h_0, m_0, \theta_0) - l(T, \delta, Z; \widehat{h}, \widehat{m}, \widehat{\theta})\right)^2\right]$$

$$= \sup_{d_{\mathsf{PF}}(\widehat{\phi}, \phi_0) \leq \varepsilon} \mathbb{E}_Z \mathbb{E}_{T, \delta | Z}\left[\log p(T, \delta, Z; h_0, m_0, \theta_0) - \log p(T, \delta, Z; \widehat{h}, \widehat{m}, \widehat{\theta})\right]^2$$

$$= 4 \sup_{d_{\mathsf{PF}}(\widehat{\phi}, \phi_0) \leq \varepsilon} \mathbb{E}_Z\left[\int \left(p(T, \delta, Z; h_0, m_0, \theta_0)\left(\log \sqrt{\frac{p(T, \delta, Z; h_0, m_0, \theta_0)}{p(T, \delta, Z; \widehat{h}, \widehat{m}, \widehat{\theta})}}\right)^2\right) \mu(dT \times d\delta)\right]$$

By Taylor's expansion on $\log x$, there exists $\xi(T, \delta, Z)$ between $p^{\frac{1}{2}}(T, \delta, Z; h_0, m_0, \theta_0)$ and $p^{\frac{1}{2}}(T, \delta, Z; \widehat{h}, \widehat{m}, \widehat{\theta})$ pointwisely such that

$$p(T, \delta, Z; h_0, m_0, \theta_0)\left(\log \sqrt{\frac{p(T, \delta, Z; h_0, m_0, \theta_0)}{p(T, \delta, Z; \widehat{h}, \widehat{m}, \widehat{\theta})}}\right)^2$$

$$= p(T, \delta, Z; h_0, m_0, \theta_0)\left(\log \sqrt{p(T, \delta, Z; h_0, m_0, \theta_0)} - \log \sqrt{p(T, \delta, Z; \widehat{h}, \widehat{m}, \widehat{\theta})}\right)^2$$

$$= \frac{p(T, \delta, Z; h_0, m_0, \theta_0)}{\xi(T, \delta, Z)^2}\left(\sqrt{p(T, \delta, Z; h_0, m_0, \theta_0)} - \sqrt{p(T, \delta, Z; \widehat{h}, \widehat{m}, \widehat{\theta})}\right)^2$$

Since

$$\frac{p(T, \delta, Z; h_0, m_0, \theta_0)}{p(T, \delta, Z; \widehat{h}, \widehat{m}, \widehat{\theta})} = e^{l(T, \delta, Z; h_0, m_0, \theta_0) - l(T, \delta, Z; \widehat{h}, \widehat{m}, \widehat{\theta})}$$

by lemma B.1, $l(T, \delta, Z; h_0, m_0, \theta_0)$ and $l(T, \delta, Z; \widehat{h}, \widehat{m}, \widehat{\theta})$ are bounded among $[0, \tau] \times \{0, 1\} \times [-1, 1]^d$ uniformly on all $\widehat{\phi} = (\widehat{h}, \widehat{m}, \widehat{\theta})$. Thus, there exist constants $C_1$ and $C_2$ such that $0 < C_1 \leq p(T, \delta, Z; h_0, m_0, \theta_0)/p(T, \delta, Z; \widehat{h}, \widehat{m}, \widehat{\theta}) \leq C_2$. This leads to the fact that

$p(T, \delta, Z; h_0, m_0, \theta_0) \frac{1}{\xi(T,\delta,Z)^2}$ is bounded. We further obtained that

$$p(T, \delta, Z; h_0, m_0, \theta_0) \left( \log \sqrt{p(T, \delta, Z; h_0, m_0, \theta_0)} - \log \sqrt{p(T, \delta, Z; \widehat{h}, \widehat{m}, \widehat{\theta})} \right)^2$$

$$\lesssim \left| \sqrt{p(T, \delta, Z; h_0, m_0, \theta_0)} - \sqrt{p(T, \delta, Z; \widehat{h}, \widehat{m}, \widehat{\theta})} \right|^2.$$

Thus, we have that

$$\sup_{d_{\mathsf{PF}}[\widehat{\phi}, \phi_0] \leq \varepsilon} \mathrm{Var}(l(T, \delta, Z; h_0, m_0, \theta_0) - l(T, \delta, Z; \widehat{h}, \widehat{m}, \widehat{\theta}))$$

$$\lesssim \sup_{d_{\mathsf{PF}}(\widehat{\phi}, \phi_0) \leq \varepsilon} \mathbb{E}_Z \left[ \int \left| \sqrt{p(T, \delta, Z; h_0, m_0, \theta_0)} - \sqrt{p(T, \delta, Z; \widehat{h}, \widehat{m}, \widehat{\theta})} \right|^2 \mu(dT \times d\delta) \right]$$

$$= \sup_{d_{\mathsf{PF}}(\widehat{\phi}, \phi_0) \leq \varepsilon} d_{\mathsf{PF}}^2 \left( \widehat{\phi}, \phi_0 \right)$$

$$\leq \varepsilon^2.$$

**Step** 3    We define that $\widetilde{\mathcal{F}}_n = \{l(T, \delta, Z; \widehat{h}, \widehat{m}, \widehat{\theta}) - l(T, \delta, Z; \pi_n h_0, \pi_n m_0, \pi_n \theta_0) : \widehat{h} \in \mathcal{H}_n, \widehat{m} \in \mathcal{M}_n, \widehat{\theta} \in \Theta\}$. Here $(\pi_n h_0, \pi_n m_0, \pi_n \theta_0)$ have been defined in lemma B.5. Obviously, we have that $\log N_{[]}(\varepsilon, \widetilde{\mathcal{F}}_n, \|\cdot\|_\infty) = \log N_{[]}(\varepsilon, \mathcal{F}_n, \|\cdot\|_\infty)$, where $\mathcal{F}$ is defined in lemma B.9. By lemma B.9, we further have that

$$\log N_{[]}(\varepsilon, \mathcal{F}_n, \|\cdot\|_\infty) \lesssim \log \frac{1}{\varepsilon} + \log N(c_h \varepsilon^{1/s_h}, \mathcal{H}_n, \|\cdot\|_2) + \log N(c_m \varepsilon^{1/s_m}, \mathcal{M}_n, \|\cdot\|_2).$$

According to [2, Theorem 7], under condition 4.3, we have that the VC-dimension of $\mathcal{H}_n$ and $\mathcal{M}_n$ satisfy that $\mathrm{VC}(\mathcal{H}_n) \lesssim n^{\frac{1}{\beta+d}} \log^3 n$ and $\mathrm{VC}(\mathcal{M}_n) \lesssim n^{\frac{d}{\beta+d}} \log^3 n$. Thus, we obtain that

$$\log N(c_h \varepsilon^{1/s_h}, \mathcal{H}_n, \|\cdot\|_2) \lesssim \frac{\mathrm{VC}(\mathcal{H}_n)}{s_h} \log \frac{1}{\varepsilon} \lesssim n^{\frac{1}{\beta+d}} \log^3 n \log \frac{1}{\varepsilon},$$

and

$$\log N(c_m \varepsilon^{1/s_m}, \mathcal{M}_n, \|\cdot\|_2) \lesssim \frac{\mathrm{VC}(\mathcal{M}_n)}{s_\nu} \log \frac{1}{\varepsilon} \lesssim n^{\frac{d}{\beta+d}} \log^3 n \log \frac{1}{\varepsilon}.$$

Thus, we obtain that $\log N_{[]}(\varepsilon, \widetilde{\mathcal{F}}_n, \|\cdot\|_\infty) \lesssim n^{\frac{d}{\beta+d}} \log^3 n \log \frac{1}{\varepsilon}$.

**Step** 4    By the Cauchy-Schwartz inequality, we have that

$$\sqrt{\mathbb{E}\left[l(T, \delta, Z; \pi_n h_0, \pi_n m_0, \pi_n \theta_0) - l(T, \delta, Z; h_0, m_0, \theta_0)\right]}$$

$$\leq \left[ \mathbb{E}(l(T, \delta, Z; \pi_n h_0, \pi_n m_0, \pi_n \theta_0) - l(T, \delta, Z; h_0, m_0, \theta_0))^2 \right]^{\frac{1}{4}}.$$

Similar to the second part and by lemma B.5, we further have that

$$\sqrt{\mathbb{E}\left[l(T, \delta, Z; \pi_n h_0, \pi_n m_0, \pi_n \theta_0) - l(T, \delta, Z; h_0, m_0, \theta_0)\right]} \lesssim \sqrt{d_{\mathsf{PF}}(\pi_n \phi_0, \phi_0)} \lesssim n^{-\frac{\beta}{2\beta+2d}}.$$

Now let

$$\tau = \frac{\beta}{2\beta + 2d} - 2 \frac{\log \log n}{\log n}$$

By Step 1,2,3 and [60, Theorem 1], we have

$$d_{\mathsf{PF}}\left(\widehat{\phi}_n, \phi_0\right) = \max \left( n^{-\tau}, d_{\mathsf{PF}}(\pi_n \phi_0, \phi_0), \right.$$

$$\left. \sqrt{\mathbb{E}\left[l(T, \delta, Z; \pi_n h_0, \pi_n m_0, \pi_n \theta_0) - l(T, \delta, Z; h_0, m_0, \theta_0)\right]} \right)$$

By lemma B.5, $d_{\mathsf{PF}}(\pi_n \phi_0, \phi_0) = O(n^{-\frac{\beta}{\beta+d}})$.

By Step 4, $\sqrt{\mathbb{E}\left[l(T, \delta, Z; \pi_n h_0, \pi_n m_0, \pi_n \theta_0) - l(T, \delta, Z; h_0, m_0, \theta_0)\right]} = O\left(n^{-\frac{\beta}{2\beta+2d}}\right)$. Thus, we have $d_{\mathsf{PF}}\left(\widehat{\phi}_n, \phi_0\right) = O(n^{-\frac{\beta}{2\beta+2d}} \log^2 n) = \widetilde{O}(n^{-\frac{\beta}{2\beta+2d}})$. □

*Proof of theorem 4.7.* The proof is divided into four steps.

**Step** 1 We denote $\psi_0 = (\nu_0, \theta_0)$ and $\widehat{\psi} = (\widehat{\nu}, \widehat{\theta})$, where $\widehat{\nu} \in \mathcal{V}_n$ and $\widehat{\theta} \in \Theta$. For arbitrary $0 < \varepsilon \leq 1$, we have that

$$\inf_{d_{\mathsf{FN}}(\widehat{\psi}, \psi_0) \geq \varepsilon} \mathbb{E}\left[ l(T, \delta, Z; \nu_0, \theta_0) - l(T, \delta, Z; \widehat{\nu}, \widehat{\theta}) \right]$$

$$= \inf_{d_{\mathsf{FN}}(\widehat{\psi}, \psi_0) \geq \varepsilon} \mathbb{E}_Z \left[ \mathbb{E}_{T, \delta|Z} \left[ \log p(T, \delta \mid Z; \nu_0, \theta_0) - \log p(T, \delta \mid Z; \widehat{\nu}, \widehat{\theta}) \right] \right]$$

$$= \inf_{d_{\mathsf{FN}}(\widehat{\psi}, \psi_0) \geq \varepsilon} \mathbb{E}_Z \left[ D_{\mathsf{KL}} \left( \mathbb{P}_{\widehat{\psi}, Z} \| \| \mathbb{P}_{\psi_0, Z} \right) \right]$$

Using the fact that $KL(\mathbb{P}_{\widehat{\psi}, Z} \| \mathbb{P}_{\psi_0, Z}) \geq 2H^2(\mathbb{P}_{\widehat{\psi}, Z} \| \mathbb{P}_{\psi_0, Z})$. Thus, we further obtain that

$$\inf_{d_{\mathsf{FN}}(\widehat{\psi}, \psi_0) \geq \varepsilon} \mathbb{E}\left[ l(T, \delta, Z; \nu_0, \theta_0) - l(T, \delta, Z; \widehat{\nu}, \widehat{\theta}) \right]$$

$$\geq \inf_{d_{\mathsf{FN}}(\widehat{\psi}, \psi_0) \geq \varepsilon} 2\mathbb{E}_Z \left[ H^2(\mathbb{P}_{\widehat{\psi}, Z} \| \mathbb{P}_{\psi_0, Z}) \right]$$

$$= 2 \inf_{d_{\mathsf{FN}}(\widehat{\psi}, \psi_0) \geq \varepsilon} d_{\mathsf{FN}}^2 \left( \widehat{\psi}, \psi_0 \right)$$

$$\geq 2\varepsilon^2.$$

**Step** 2 We consider the following derivation.

$$\sup_{d_{\mathsf{FN}}(\widehat{\psi}, \psi_0) \leq \varepsilon} \mathrm{Var} \left[ l(T, \delta, Z; \nu_0, \theta_0) - l(T, \delta, Z; \widehat{\nu}, \widehat{\theta}) \right]$$

$$\leq \sup_{d_{\mathsf{FN}}(\widehat{\psi}, \psi_0) \leq \varepsilon} \mathbb{E}\left[ \left( l(T, \delta, Z; \nu_0, \theta_0) - l(T, \delta, Z; \widehat{\nu}, \widehat{\theta}) \right)^2 \right]$$

$$= \sup_{d_{\mathsf{FN}}(\widehat{\psi}, \psi_0) \leq \varepsilon} \mathbb{E}_Z \left[ \mathbb{E}_{T, \delta|Z} \left[ \left( \log p(T, \delta, Z; \nu_0, \theta_0) - \log p(T, \delta, Z; \widehat{\nu}, \widehat{\theta}) \right)^2 \right] \right]$$

$$= 4 \sup_{d_{\mathsf{FN}}(\widehat{\psi}, \psi_0) \leq \varepsilon} \mathbb{E}_Z \left[ \int \left( p(T, \delta, Z; \nu_0, \theta_0) (\log \sqrt{\frac{p(T, \delta, Z; \nu_0, \theta_0)}{p(T, \delta, Z; \widehat{\nu}, \widehat{\theta})}})^2 \right) \mu(dT \times d\delta) \right]$$

By Taylor's expansion on $\log x$, there exists $\eta(T, \delta, Z)$ between $\sqrt{p(T, \delta, Z; \nu_0, \theta_0)}$ and $\sqrt{p(T, \delta, Z; \widehat{\nu}, \widehat{\theta})}$ pointwisely such that

$$p(T, \delta, Z; \nu_0, \theta_0) (\log \sqrt{\frac{p(T, \delta, Z; \nu_0, \theta_0)}{p(T, \delta, Z; \widehat{\nu}, \widehat{\theta})}})^2$$

$$= p(T, \delta, Z; \nu_0, \theta_0) \left( \log \sqrt{p(T, \delta, Z; \nu_0, \theta_0)} - \log \sqrt{p(T, \delta, Z; \widehat{\nu}, \widehat{\theta})} \right)^2$$

$$= \frac{p(T, \delta, Z; \nu_0, \theta_0)}{\eta(T, \delta, Z)^2} \left( \sqrt{p(T, \delta, Z; \nu_0, \theta_0)} - \sqrt{p(T, \delta, Z; \widehat{\nu}, \widehat{\theta})} \right)^2$$

Since $p(T, \delta, Z; \nu_0, \theta_0)/p(T, \delta, Z; \widehat{\nu}, \widehat{\theta}) = e^{l(T, \delta, Z; \nu_0, \theta_0) - l(T, \delta, Z; \widehat{\nu}, \widehat{\theta})}$, by lemma B.2, $l(T, \delta, Z; \nu_0, \theta_0)$ and $l(T, \delta, Z; \widehat{\nu}, \widehat{\theta})$ are bounded on $[0, \tau] \times \{0, 1\} \times [-1, 1]^d$ uniformly for all $\widehat{\psi} = (\widehat{\nu}, \widehat{\theta})$. Thus there exist constants $C_3$ and $C_4$ such that $0 < C_3 \leq p(T, \delta, Z; \nu_0, \theta_0)/p(T, \delta, Z; \widehat{\nu}, \widehat{\theta}) \leq C_4$. This leads to the fact that $p(T, \delta, Z; \nu_0, \theta_0) \frac{1}{\eta(T, \delta, Z)^2}$ is bounded. We further have that

$$p(T, \delta, Z; \nu_0, \theta_0) \left( \log \sqrt{p(T, \delta, Z; \nu_0, \theta_0)} - \log \sqrt{p(T, \delta, Z; \widehat{\nu}, \widehat{\theta})} \right)^2$$

$$\lesssim \left| \sqrt{p(T, \delta, Z; \nu_0, \theta_0)} - \sqrt{p(T, \delta, Z; \widehat{\nu}, \widehat{\theta})} \right|^2.$$

Thus, we have that

$$\sup_{d_{\mathsf{FN}}(\widehat{\psi},\psi_0)\le\varepsilon} \mathrm{Var}\left[l(T,\delta,Z;\nu_0,\theta_0)-l(T,\delta,Z;\widehat{\nu},\widehat{\theta})\right]$$

$$\lesssim \sup_{d_{\mathsf{FN}}(\widehat{\psi},\psi_0)\le\varepsilon} \mathbb{E}_Z\left[\int\left|\sqrt{p(T,\delta,Z;\nu_0,\theta_0)}-\sqrt{p(T,\delta,Z;\widehat{\nu},\widehat{\theta})}\right|^2\mu(dT\times d\delta)\right]$$

$$= \sup_{d_{\mathsf{FN}}(\widehat{\psi},\psi_0)\le\varepsilon} d_{\mathsf{FN}}^2\left(\widehat{\psi},\psi_0\right)$$

$$\le \varepsilon^2.$$

**Step** 3 We define that $\widetilde{\mathcal{G}}_n = \{l(T,\delta,Z;\widehat{\nu},\widehat{\theta})-l(T,\delta,Z;\pi_n\nu_0,\pi_n\theta_0):\widehat{\nu}\in\mathcal{V}_n,\theta\in\Theta\}$. Here $(\pi_n\nu_0,\pi_n\theta_0)$ have been defined in lemma B.6. Obviously, we have that $\log N_{[]}(\varepsilon,\widetilde{\mathcal{G}}_n,\|\cdot\|_\infty) = \log N_{[]}(\varepsilon,\mathcal{G}_n,\|\cdot\|_\infty)$, where $\mathcal{G}$ is defined in lemma B.10. By lemma B.10, we further obtain that

$$\log N_{[]}(\varepsilon,\mathcal{G}_n,\|\cdot\|_\infty)\lesssim\log\frac{1}{\varepsilon}+\log N(c_\nu\varepsilon^{1/s_\nu},\mathcal{V}_n,\|\cdot\|_2).$$

According to [2, Theorem 7], under condition 4.4, we have that the VC-dimension of $\mathcal{V}_n$ satisfies that $\mathrm{VC}(\mathcal{V}_n)\lesssim n^{\frac{d+1}{\beta+d+1}}\log^3 n$. Thus, we obtain that

$$\log N(c_h\varepsilon^{1/s_\nu},\mathcal{V}_n,\|\cdot\|_2)\lesssim\frac{\mathrm{VC}(\mathcal{V}_n)}{s_\nu}\log\frac{1}{\varepsilon}\lesssim n^{\frac{d+1}{\beta+d+1}}\log^3 n\log\frac{1}{\varepsilon}.$$

Furthermore, we get that $\log N_{[]}(\varepsilon,\widetilde{\mathcal{G}}_n,\|\cdot\|_\infty)\lesssim n^{\frac{d+1}{\beta+d+1}}\log^3 n\log\frac{1}{\varepsilon}$.

**Step** 4 By the Cauchy-Schwartz inequality, we have that

$$\sqrt{\mathbb{E}[l(T,\delta,Z;\pi_n\nu_0,\pi_n\theta_0)-l(T,\delta,Z;\nu_0,\theta_0)]}\le\left[\mathbb{E}\left(l(T,\delta,Z;\pi_n\nu_0,\pi_n\theta_0)-l(T,\delta,Z;\nu_0,\theta_0)\right)^2\right]^{\frac{1}{4}}.$$

Similar to the second part and by lemma B.6, we further obtain that

$$\sqrt{\mathbb{E}[l(T,\delta,Z;\pi_n\nu_0,\pi_n\theta_0)-l(T,\delta,Z;\nu_0,\theta_0)]}\lesssim\sqrt{d_{\mathsf{FN}}(\pi_n\psi_0,\psi_0)}\lesssim n^{-\frac{\beta}{2\beta+2d+2}}$$

Now let

$$\tau=\frac{\beta}{2\beta+2d+2}-2\frac{\log\log n}{\log n}.$$

By step 1,2,3 and Step 1,2,3 and [60, Theorem 1],

$$d_{\mathsf{FN}}\left(\widehat{\psi}_n,\psi_0\right)=\max\left(n^{-\tau},d_{\mathsf{FN}}(\pi_n\psi_0,\psi_0),\right.$$

$$\left.\sqrt{\mathbb{E}[l(T,\delta,Z;\pi_n\nu_0,\pi_n\theta_0)-l(T,\delta,Z;\nu_0,\theta_0)]}\right)$$

By lemmaB.6, $d_{\mathsf{FN}}(\pi_n\psi_0,\psi_0)=O(n^{-\frac{\beta}{\beta+d+1}})$

By Step 4, $\sqrt{\mathbb{E}[l(T,\delta,Z;\pi_n\nu_0,\pi_n\theta_0)-l(T,\delta,Z;\nu_0,\theta_0)]}=O(n^{-\frac{\beta}{2\beta+2d+2}})$. Thus, we have $d_{\mathsf{FN}}\left(\widehat{\psi}_n,\psi_0\right)=O(n^{-\frac{\beta}{2\beta+2d+2}}\log^2 n)=\widetilde{O}(n^{-\frac{\beta}{2\beta+2d+2}})$. □

### B.4 Proofs of technical lemmas

*Proof of lemma B.1.* Since $h_0(T)\in\mathcal{W}_M^\beta([0,\tau])$ and $m_0(Z)\in\mathcal{W}_M^\beta([-1,1]^d)$, we have that $h_0(T)\le M$, $m_0(Z)\le M$ and $e^{m_0(Z)}\int_0^T h_0(s)ds\le\tau e^{2M}$. Let $\mathcal{B}=[0,\tau e^{2M}]$, we have that

$$|l(T,\delta,Z;h_0,m_0,\theta_0)|$$

$$\le\left|\log g_{\theta_0}\left(e^{m_0(Z)}\int_0^T e^{h_0(s)}ds\right)\right|+|h_0(T)|+|m_0(Z)|+\left|G_{\theta_0}\left(e^{m_0(Z)}\int_0^T e^{h_0(s)}ds\right)\right|$$

$$\le 2M+\sup_{x\in\mathcal{B}}|\log g_{\theta_0}(x)|+\sup_{x\in\mathcal{B}}|G_{\theta_0}(x)|$$

By condition 5, we have that $l(T,\delta,Z;h_0,m_0,\theta_0)$ is bounded for $(T,\delta,Z)\in[0,\tau]\times\{0,1\}\times[-1,1]^d$. The proof of the boundness of $l(T,\delta,Z;\widehat{h},\widehat{m},\widehat{\theta})$ is similar. □

*Proof of lemma B.2.* Since $\nu_0(T,Z) \in \mathcal{W}_M^\beta([0,\tau] \times [-1,1]^d)$, we have $\nu_0(T,Z) \le M$ and $\int_0^T e^{\nu(s,Z)}ds \le \tau e^M$. Let $\mathcal{B} = [0, \tau e^M]$, we have that

$$|l(T,\delta,Z;\nu_0,\theta_0)|$$
$$\le \left|\log G'_{\theta_0}\left(\int_0^T e^{\nu_0(s,Z)}ds\right)\right| + |\nu_0(T,Z)| + \left|G_{\theta_0}\left(\int_0^T e^{\nu_0(s,Z)}ds\right)\right|$$
$$\le M + \sup_{x\in\mathcal{B}}\left|\log G'_{\theta_0}(x)\right| + \sup_{x\in\mathcal{B}}|G_{\theta_0}(x)|.$$

By condition 4.5, we have that $l(T,\delta,Z;\nu_0,\theta_0)$ is bounded among $(T,\delta,Z) \in [0,\tau] \times \{0,1\} \times [-1,1]^d$ The proof of the boundedness of $l(T,\delta,Z;\widehat{\nu},\widehat{\theta})$ is similar. $\square$

*Proof of lemma B.3.* By definition, we have that

$$|l(T,\delta,Z;h_0,m_0,\theta_0) - l(T,\delta,Z;\widehat{h},\widehat{m},\widehat{\theta})|$$
$$\le \left|\log g_{\theta_0}\left(e^{m_0(Z)}\int_0^T e^{h_0(s)}ds\right) - \log g_{\widehat{\theta}}\left(e^{\widehat{m}(Z)}\int_0^T e^{\widehat{h}(s)}ds\right)\right| + \left|h_0(T) - \widehat{h}(T)\right|$$
$$+ |m_0(Z) - \widehat{m}(Z)| + \left|G_{\theta_0}\left(e^{m_0(Z)}\int_0^T e^{h_0(s)}ds\right) - G_{\widehat{\theta}}\left(e^{\widehat{m}(Z)}\int_0^T e^{\widehat{h}(s)}ds\right)\right|.$$

Let $\mathcal{B} = [0, \tau\max(e^{2M}, e^{M_h+M_m})]$. By Taylor's expansion, we can further show that

$$|l(T,\delta,Z;h_0,m_0,\theta_0) - l(T,\delta,Z;\widehat{h},\widehat{m},\widehat{\theta})|$$
$$\le \sup_{\tilde\theta\in\Theta,\tilde x\in\mathcal{B}}\left|\frac{\partial \log g_{\tilde\theta}(\tilde x)}{\partial\tilde\theta}\right| \cdot \left|\theta_0 - \widehat{\theta}\right|$$
$$+ \sup_{\tilde\theta\in\Theta,\tilde x\in\mathcal{B}}\left|\frac{\partial \log g_{\tilde\theta}(\tilde x)}{\partial\tilde x}\right| \cdot \left|e^{m_0(Z)}\int_0^T e^{h_0(s)}ds - e^{\widehat{m}(Z)}\int_0^T e^{\widehat{h}(s)}ds\right|$$
$$+ |h_0(T) - \widehat{h}(T)| + |m_0(Z) - \widehat{m}(Z)| + \sup_{\tilde\theta\in\Theta,\tilde x\in\mathcal{B}}\left|\frac{\partial G_{\tilde\theta}(\tilde x)}{\partial\tilde\theta}\right| \cdot \left|\theta_0 - \widehat{\theta}\right|$$
$$+ \sup_{\tilde\theta\in\Theta,\tilde x\in\mathcal{B}}\left|\frac{\partial G_{\tilde\theta}(\tilde x)}{\partial\tilde x}\right| \cdot \left|e^{m_0(Z)}\int_0^T e^{h_0(s)}ds - e^{\widehat{m}(Z)}\int_0^T e^{\widehat{h}(s)}ds\right|.$$

Again, by Taylor's expansion, we have that

$$\left|e^{m_0(Z)}\int_0^T e^{h_0(s)}ds - e^{\widehat{m}(Z)}\int_0^T e^{\widehat{h}(s)}ds\right|$$
$$\le \left|e^{m_0(Z)}\int_0^T (e^{h_0(s)} - e^{\widehat{h}(s)})ds\right| + \left|(e^{m_0(Z)} - e^{\widehat{m}(Z)})\int_0^T e^{\widehat{h}(s)}ds\right|$$
$$\le e^M \cdot \tau e^{\max(M,M_h)}\left\|h_0 - \widehat{h}\right\|_\infty + \tau e^{M_h} \cdot e^{\max(M,M_m)}\|m_0 - \widehat{m}\|_\infty.$$

Finally, we obtain that

$$|l(T,\delta,Z;h_0,m_0,\theta_0) - l(T,\delta,Z;\widehat{h},\widehat{m},\widehat{\theta})|$$
$$\le \sup_{\tilde\theta\in\Theta,\tilde x\in\mathcal{B}}\left|\frac{\partial \log g_{\tilde\theta}(\tilde x)}{\partial\tilde x}\right| \cdot \left[e^M \cdot \tau e^{\max(M,M_h)}\|h_0 - \widehat{h}\|_\infty + \tau e^{M_h} \cdot e^{\max(M,M_m)}\|m_0 - \widehat{m}\|_\infty\right]$$
$$+ \sup_{\tilde\theta\in\Theta,\tilde x\in\mathcal{B}}\left|\frac{\partial \log g_{\tilde\theta}(\tilde x)}{\partial\tilde\theta}\right| \cdot \left|\theta_0 - \widehat{\theta}\right| + \left|h_0(T) - \widehat{h}(T)\right| + |m_0(Z) - \widehat{m}(Z)|$$
$$+ \sup_{\tilde\theta\in\Theta,\tilde x\in\mathcal{B}}\left|\frac{\partial G_{\tilde\theta}(\tilde x)}{\partial\tilde\theta}\right| \cdot \left|\theta_0 - \widehat{\theta}\right|$$
$$+ \sup_{\tilde\theta\in\Theta,\tilde x\in\mathcal{B}}\left|\frac{\partial G_{\tilde\theta}(\tilde x)}{\partial\tilde x}\right| \cdot \left[e^M \cdot \tau e^{\max(M,M_h)}\|h_0 - \widehat{h}\|_\infty + \tau e^{M_h} \cdot e^{\max(M,M_m)}\|m_0 - \widehat{m}\|_\infty\right].$$

Taking supremum on both sides, we conclude that

$$\|l(T, \delta, Z; h_0, m_0, \theta_0) - l(T, \delta, Z; \widehat{h}, \widehat{m}, \widehat{\theta})\|_\infty \lesssim |\theta_0 - \widehat{\theta}| + \|h_0 - \widehat{h}\|_\infty + \|m_0 - \widehat{m}\|_\infty.$$

The proof of the second inequality is similar. □

*Proof of lemma B.4.* By definition, we have that

$$\begin{aligned}
&|l(T, \delta, Z; \nu_0, \theta_0) - l(T, \delta, Z; \widehat{\nu}, \widehat{\theta})| \\
&\leq \left|\log g_{\theta_0}\left(\int_0^T e^{\nu_0(s,Z)} ds\right) - \log g_{\widehat{\theta}}\left(\int_0^T e^{\widehat{\nu}(s,Z)} ds\right)\right| + |\nu_0(T, Z) - \widehat{\nu}(T, Z)| \\
&\quad + \left|G_{\theta_0}\left(\int_0^T e^{\nu_0(s,Z)} ds\right) - G_{\widehat{\theta}}\left(\int_0^T e^{\widehat{\nu}(s,Z)} ds\right)\right|.
\end{aligned}$$

Let $\mathcal{B} = [0, \tau \max(e^M, e^{M_\nu})]$. By Taylor's expansion, we can further show that

$$\begin{aligned}
&|l(T, \delta, Z; \nu_0, \theta_0) - l(T, \delta, Z; \widehat{\nu}, \widehat{\theta})| \\
&\leq \sup_{\tilde{\theta} \in \Theta, \tilde{x} \in \mathcal{B}} \left|\frac{\partial \log g_{\tilde{\theta}}(\tilde{x})}{\partial \tilde{\theta}}\right| \cdot \left|\theta_0 - \widehat{\theta}\right| + \sup_{\tilde{\theta} \in \Theta, \tilde{x} \in \mathcal{B}} \left|\frac{\partial \log g_{\tilde{\theta}}(\tilde{x})}{\partial \tilde{x}}\right| \cdot \left|\int_0^T e^{\nu_0(s,Z)} ds - \int_0^T e^{\widehat{\nu}(s,Z)} ds\right| \\
&\quad + |\nu_0(T, Z) - \widehat{\nu}(T, Z)| + \sup_{\tilde{\theta} \in \Theta, \tilde{x} \in \mathcal{B}} \left|\frac{\partial G_{\tilde{\theta}}(\tilde{x})}{\partial \tilde{\theta}}\right| \cdot |\theta_0 - \widehat{\theta}| \\
&\quad + \sup_{\tilde{\theta} \in \Theta, \tilde{x} \in \mathcal{B}} \left|\frac{\partial G_{\tilde{\theta}}(\tilde{x})}{\partial \tilde{x}}\right| \cdot \left|\int_0^T e^{\nu_0(s,Z)} ds - \int_0^T e^{\widehat{\nu}(s,Z)} ds\right|.
\end{aligned}$$

Again, by Taylor's expansion,

$$\left|\int_0^T e^{\nu_0(s,Z)} ds - \int_0^T e^{\widehat{\nu}(s,Z)} ds\right| \leq \tau e^{\max(M, M_\nu)} \|\nu_0 - \widehat{\nu}\|_\infty,$$

Finally, we obtain that

$$\begin{aligned}
&\left|l(T, \delta, Z; \nu_0, \theta_0) - l(T, \delta, Z; \widehat{\nu}, \widehat{\theta})\right| \\
&\leq \sup_{\tilde{\theta} \in \Theta, \tilde{x} \in \mathcal{B}} \left|\frac{\partial \log g_{\tilde{\theta}}(\tilde{x})}{\partial \tilde{\theta}}\right| \cdot \left|\theta_0 - \widehat{\theta}\right| + \sup_{\tilde{\theta} \in \Theta, \tilde{x} \in \mathcal{B}} \left|\frac{\partial \log g_{\tilde{\theta}}(\tilde{x})}{\partial \tilde{x}}\right| \cdot \tau e^{\max(M, M_\nu)} \|\nu_0 - \widehat{\nu}\|_\infty \\
&\quad + |\nu_0(T, Z) - \widehat{\nu}(T, Z)| + \sup_{\tilde{\theta} \in \Theta, \tilde{x} \in \mathcal{B}} \left|\frac{\partial G_{\tilde{\theta}}(\tilde{x})}{\partial \tilde{\theta}}\right| \cdot \left|\theta_0 - \widehat{\theta}\right| \\
&\quad + \sup_{\tilde{\theta} \in \Theta, \tilde{x} \in \mathcal{B}} \left|\frac{\partial G_{\tilde{\theta}}(\tilde{x})}{\partial \tilde{x}}\right| \cdot \tau e^{\max(M, M_\nu)} \|\nu_0 - \widehat{\nu}\|_\infty.
\end{aligned}$$

Taking supremum on both sides, we conclude that

$$\|l(T, \delta, Z; \nu_0, \theta_0) - l(T, \delta, Z; \widehat{\nu}, \widehat{\theta})\|_\infty \lesssim |\theta_0 - \widehat{\theta}| + \|\nu_0 - \widehat{\nu}\|_\infty,$$

The proof of the second inequality is similar. □

*Proof of lemma B.5.* According to [72, Theorem 1], there exist approximating functions $\widehat{h}^*$ and $\widehat{m}^*$ such that $\|\widehat{h}^* - h_0\|_\infty = O\left(n^{-\frac{\beta}{\beta+d}}\right)$ and $\|\widehat{m}^* - m_0\|_\infty = O\left(n^{-\frac{\beta}{\beta+d}}\right)$. Let $\pi_n h_0 = \widehat{h}^*$,

$\pi_n m_0 = \widehat{m}^*$, and $\pi_n \theta = \theta_0$. We have that

$$d_{\mathsf{PF}}\left(\pi_n \phi_0, \phi_0\right)$$

$$= \sqrt{\mathbb{E}_Z\left[\int |\sqrt{p(T, \delta \mid Z; \pi_n h_0, \pi_n m_0, \pi_n \theta_0)} - \sqrt{p(T, \delta \mid Z; h_0, m_0, \theta_0)}|^2 \mu(dT \times d\delta)\right]}$$

$$= \sqrt{\mathbb{E}_Z\left[\int [e^{\frac{1}{2}l(T, \delta, Z; \pi_n h_0, \pi_n m_0, \pi_n \theta_0)} - e^{\frac{1}{2}l(T, \delta, Z; h_0, m_0, \theta_0)}]^2 f_{C|Z}(T)^{1-\delta} S_{C|Z}(T)^\delta \mu(dT \times d\delta)\right]}$$

$$\leq \left\| e^{\frac{1}{2}l(T, \delta, Z; \pi_n h_0, \pi_n m_0, \pi_n \theta_0)} - e^{\frac{1}{2}l(T, \delta, Z; h_0, m_0, \theta_0)} \right\|_\infty$$

$$\times \sqrt{\mathbb{E}_Z\left[\int f_{C|Z}(T)^{1-\delta} S_{C|Z}(T)^\delta \mu(dT \times d\delta)\right]}.$$

By lemma B.1 and B.3, we have that

$$\|e^{\frac{1}{2}l(T, \delta, Z; \pi_n h_0, \pi_n m_0, \pi_n \theta_0)} - e^{\frac{1}{2}l(T, \delta, Z; h_0, m_0, \theta_0)}\|_\infty$$
$$\lesssim \|\pi_n \theta_0 - \theta_0\| + \|\pi_n h_0 - h_0\|_\infty + \|\pi_n m_0 - m_0\|_\infty$$
$$= O\left(n^{-\frac{\beta}{\beta+d}}\right).$$

Since $f_{C|Z}(T)^{1-\delta} \leq f_{C|Z}(T) + 1$ and $S_{C|Z}(T)^\delta \leq 1$, we also have that

$$\sqrt{\mathbb{E}_Z\left[\int f_{C|Z}(T)^{1-\delta} S_{C|Z}(T)^\delta \mu(dT \times d\delta)\right]} \leq \sqrt{\mathbb{E}_Z\left[\int (1 + f_{C|Z}(T))\mu(dT \times d\delta)\right]}$$
$$\leq \sqrt{2 + 2\tau}.$$

Thus, we obtain that $d_{\mathsf{PF}}\left(\pi_n \phi_0, \phi_0\right) = O\left(n^{-\frac{\beta}{\beta+d}}\right)$. $\qquad \square$

*Proof of lemma B.6.* According to [72, Theorem 1], there exists an approximating function $\widehat{\nu}^*$ such that $\|\widehat{\nu}^* - \nu_0\|_\infty = O\left(n^{-\frac{\beta}{\beta+d+1}}\right)$. Let $\pi_n \nu_0 = \widehat{\nu}^*$ and $\pi_n \theta_0 = \theta_0$. We have that

$$d_{\mathsf{FN}}\left(\pi_n \psi_0, \psi_0\right)$$

$$= \sqrt{\mathbb{E}_Z\left[\int \left|\sqrt{p(T, \delta \mid Z; \pi_n \nu_0, \pi_n \theta_0)} - \sqrt{p(T, \delta \mid Z; \nu_0, \theta_0)}\right|^2 \mu(dT \times d\delta)\right]}$$

$$= \sqrt{\mathbb{E}_Z\left[\int \left[e^{\frac{1}{2}l(T, \delta, Z; \pi_n \nu_0, \pi_n \theta_0)} - e^{\frac{1}{2}l(T, \delta, Z; \nu_0, \theta_0)}\right]^2 f_{C|Z}(T)^{1-\delta} S_{C|Z}(T)^\delta \mu(dT \times d\delta)\right]}$$

$$\leq \left\| \frac{1}{2}e^{l(T, \delta, Z; \pi_n \nu_0, \pi_n \theta_0)} - \frac{1}{2}e^{l(T, \delta, Z; \nu_0, \theta_0)} \right\|_\infty \sqrt{\mathbb{E}_Z\left[\int f_{C|Z}(T)^{1-\delta} S_{C|Z}(T)^\delta \mu(dT \times d\delta)\right]}.$$

By lemma B.2 and B.4, we have that

$$\left\| e^{\frac{1}{2}l(T, \delta, Z; \pi_n \nu_0, \pi_n \theta_0)} - e^{\frac{1}{2}l(T, \delta, Z; \nu_0, \theta_0)} \right\|_\infty \lesssim \|\pi_n \theta_0 - \theta_0\| + \|\pi_n \nu_0 - \nu_0\|_\infty$$
$$= O\left(n^{-\frac{\beta}{\beta+d+1}}\right).$$

Since $f_{C|Z}(T)^{1-\delta} \leq f_{C|Z}(T) + 1$ and $S_{C|Z}(T)^\delta \leq 1$, we also have that

$$\sqrt{\mathbb{E}_Z\left[\int f_{C|Z}(T)^{1-\delta} S_{C|Z}(T)^\delta \mu(dT \times d\delta)\right]} \leq \sqrt{\mathbb{E}_Z\left[\int (1 + f_{C|Z}(T))\mu(dT \times d\delta)\right]}$$
$$\leq \sqrt{2 + 2\tau}.$$

Thus, we obtain that $d_{\mathsf{FN}}\left(\pi_n \psi_0, \psi_0\right) = O\left(n^{-\frac{\beta}{\beta+d+1}}\right)$. $\qquad \square$

*Proof of lemma B.7.* The left inequality is trivial according to the definition of covering number. We need to show that the correctness of the right inequality.

Suppose that we have $\{B(g_i, \frac{\varepsilon}{2})\}, i = 1 \ldots, N$, where $N = N(\frac{\varepsilon}{2}, \mathcal{F}, \|\cdot\|)$, are the minimal number of $\frac{\varepsilon}{2}$-ball that covers $\mathcal{F}$. Then there exists at least one $f_i \in \mathcal{F}$ such that $f_i \in B(g_i, \varepsilon)$. Consider the following $\varepsilon - balls$ $\{B(f_i, \varepsilon)\}, i = 1 \ldots, N$. For arbitrary $f \in \mathcal{F} \cap B(g_i, \frac{\varepsilon}{2})$, we have that $\|f - f_i\| \leq \|f - g_i\| + \|f_i - g_i\| \leq \varepsilon$. Thus $\{B(f_i, \varepsilon)\}, i = 1 \ldots, N$ forms a $\varepsilon$-covering of $\mathcal{F}$. By definition, we have that $\widetilde{N}(\varepsilon, \mathcal{F}, \|\cdot\|) \leq N(\frac{\varepsilon}{2}, \mathcal{F}, \|\cdot\|)$. $\qquad \square$

*Proof of lemma B.8.* The proof of the first two inequalities follows exactly the same steps of lemma B.7. Here we just need to mention the rest of the statement that $\widetilde{N}_{[]}(\varepsilon, \mathcal{F}, \|\cdot\|_\infty) = \widetilde{N}(\frac{\varepsilon}{2}, \mathcal{F}, \|\cdot\|_\infty)$. We first choose a set of $\frac{\varepsilon}{2}$-covering balls $\{B(f_i, \frac{\varepsilon}{2})\}, i = 1, \ldots, N_1$, where $N_1 = \widetilde{N}(\frac{\varepsilon}{2}, \mathcal{F}, \|\cdot\|_\infty)$. Now we construct a set of brackets $\{[l_i, u_i]\}, i = 1 \ldots, N_1$, where $l_i = f_i - \frac{\varepsilon}{2}$ and $u_i = f_i + \frac{\varepsilon}{2}$. Noting that the bracket $\{[l_i, u_i]\}$ is exactly the same as $B(f_i, \frac{\varepsilon}{2})$, The set $\{[l_i, u_i]\}, i = 1, \ldots, N_1$ covers $\mathcal{F}$, which leads to $\widetilde{N}_{[]}(\varepsilon, \mathcal{F}, \|\cdot\|_\infty) \leq \widetilde{N}(\frac{\varepsilon}{2}, \mathcal{F}, \|\cdot\|_\infty)$. Likewise, we have that $\widetilde{N}_{[]}(\varepsilon, \mathcal{F}, \|\cdot\|_\infty) \geq \widetilde{N}(\frac{\varepsilon}{2}, \mathcal{F}, \|\cdot\|_\infty)$. Consequently, we have that $\widetilde{N}_{[]}(\varepsilon, \mathcal{F}, \|\cdot\|_\infty) = \widetilde{N}(\frac{\varepsilon}{2}, \mathcal{F}, \|\cdot\|_\infty)$. $\qquad \square$

*Proof of lemma B.9.* By lemma B.8, first we have that $N_{[]}(\varepsilon, \mathcal{F}_n, \|\cdot\|_\infty) \leq \widetilde{N}_{[]}(\varepsilon, \mathcal{F}_n, \|\cdot\|_\infty)$. By lemma B.3, there exists a constant $c_1 > 0$ such that for arbitrary $\widehat{h}_1, \widehat{h}_2 \in \mathcal{H}_n, \widehat{m}_1, \widehat{m}_2 \in \mathcal{M}_n$ and $\widehat{\theta}_1, \widehat{\theta}_2 \in \Theta$, we have that

$$\|l(T, \delta, Z; \widehat{h}_1, \widehat{m}_1, \theta_1) - l(T, \delta, Z; \widehat{h}_2, \widehat{m}_2, \theta_2)\|_\infty \leq c_1[|\widehat{\theta}_1 - \widehat{\theta}_2| + \|\widehat{h}_1 - \widehat{h}_2\|_\infty + \|\widehat{m}_1 - \widehat{m}_2\|_\infty],$$

which indicates that as long as $|\widehat{\theta}_1 - \widehat{\theta}_2| \leq \frac{\varepsilon}{3c_1}$, $\|\widehat{h}_1 - \widehat{h}_2\|_\infty \leq \frac{\varepsilon}{3c_1}$ and $\|\widehat{m}_1 - \widehat{m}_2\|_\infty \leq \frac{\varepsilon}{3c_1}$, we have that $\|l(T, \delta, Z; \widehat{h}_1, \widehat{m}_1, \theta_1) - l(T, \delta, Z; \widehat{h}_2, \widehat{m}_2, \theta_2)\|_\infty \leq \varepsilon$. Consequently, we have that

$$\widetilde{N}_{[]}(\varepsilon, \mathcal{F}_n, \|\cdot\|_\infty) \leq \widetilde{N}_{[]}(\frac{\varepsilon}{3c_1}, \Theta, \|\cdot\|_\infty) \times \widetilde{N}_{[]}(\frac{\varepsilon}{3c_1}, \mathcal{H}_n, \|\cdot\|_\infty) \times \widetilde{N}_{[]}(\frac{\varepsilon}{3c_1}, \mathcal{M}_n, \|\cdot\|_\infty).$$

Since $\Theta$ is a compact set on $\mathbb{R}$, by lemma B.8 and traditional volume argument, we have that $\widetilde{N}_{[]}(\frac{\varepsilon}{3c_1}, \Theta, \|\cdot\|_\infty) \leq N_{[]}(\frac{\varepsilon}{6c_1}, \Theta, \|\cdot\|_\infty) \lesssim \frac{1}{\varepsilon}$.

For $\widetilde{N}_{[]}(\frac{\varepsilon}{3c_1}, \mathcal{H}_n, \|\cdot\|_\infty)$, by lemma B.8, we have that $\widetilde{N}_{[]}(\frac{\varepsilon}{3c_1}, \mathcal{H}_n, \|\cdot\|_\infty) = \widetilde{N}(\frac{\varepsilon}{3c_1}, \mathcal{H}_n, \|\cdot\|_\infty)$. By [12, Lemma 2], there exists a constant $c_2 > 0$ such that $\|\widehat{h}_1 - \widehat{h}_2\|_\infty \leq c_2\|\widehat{h}_1 - \widehat{h}_2\|_2^{s_h}$, which leads to $\widetilde{N}(\frac{\varepsilon}{3c_1}, \mathcal{H}_n, \|\cdot\|_\infty) \leq \widetilde{N}(\frac{\varepsilon^{1/s_h}}{(3c_1 c_2)^{1/s_h}}, \mathcal{H}_n, \|\cdot\|_2)$. By lemma B.7 we further have that $\widetilde{N}(\frac{\varepsilon^{1/s_h}}{(3c_1 c_2)^{1/s_h}}, \mathcal{H}_n, \|\cdot\|_2) \leq N(\frac{\varepsilon^{1/s_h}}{2(3c_1 c_2)^{1/s_h}}, \mathcal{H}_n, \|\cdot\|_2)$. Let $c_h = \frac{1}{2(3c_1 c_2)^{1/s_h}}$. We have that $\widetilde{N}_{[]}(\frac{\varepsilon}{3c_1}, \mathcal{H}_n, \|\cdot\|_\infty) \leq N(c_h \varepsilon^{1/s_h}, \mathcal{H}_n, \|\cdot\|_2)$.

Similarly, there exists a constant $c_m > 0$ such that $\widetilde{N}_{[]}(\frac{\varepsilon}{3c_1}, \mathcal{M}_n, \|\cdot\|_\infty) \leq N(c_m \varepsilon^{1/s_m}, \mathcal{M}_n, \|\cdot\|_2)$. Thus, finally we can obtain that

$$N_{[]}(\varepsilon, \mathcal{F}_n, \|\cdot\|_\infty) \lesssim \frac{1}{\varepsilon} N(c_h \varepsilon^{1/s_h}, \mathcal{H}_n, \|\cdot\|_2) \times N(c_m \varepsilon^{1/s_m}, \mathcal{M}_n, \|\cdot\|_2).$$

$\qquad \square$

*Proof of lemma B.10.* By lemma B.8, first we have $N_{[]}(\varepsilon, \mathcal{G}_n, \|\cdot\|_\infty) \leq \widetilde{N}_{[]}(\varepsilon, \mathcal{G}_n, \|\cdot\|_\infty)$. By lemma B.4, there exists a constant $c_3 > 0$ such that for arbitrary $\widehat{\nu}_1, \widehat{\nu}_2 \in \mathcal{V}_n$ and $\widehat{\theta}_1, \widehat{\theta}_2 \in \Theta$, we have that

$$\|l(T, \delta, Z; \widehat{\nu}_1, \widehat{\theta}_1) - l(T, \delta, Z; \widehat{\nu}_2, \widehat{\theta}_2)\|_\infty \leq c_3[|\widehat{\theta}_1 - \widehat{\theta}_2| + \|\widehat{\nu}_1 - \widehat{\nu}_2\|_\infty],$$

which indicates that as long as $|\widehat{\theta}_1 - \widehat{\theta}_2| \leq \frac{\varepsilon}{2c_3}$ and $\|\widehat{\nu}_1 - \widehat{\nu}_2\|_\infty \leq \frac{\varepsilon}{2c_3}$, we have that $\|l(T, \delta, Z; \widehat{\nu}_1, \widehat{\theta}_1) - l(T, \delta, Z; \widehat{\nu}_2, \widehat{\theta}_2)\|_\infty \leq \varepsilon$. Thus, we have:

$$\widetilde{N}_{[]}(\varepsilon, \mathcal{G}_n, \|\cdot\|_\infty) \leq \widetilde{N}_{[]}(\frac{\varepsilon}{2c_3}, \Theta, \|\cdot\|_\infty) \times \widetilde{N}_{[]}(\frac{\varepsilon}{2c_3}, \mathcal{V}_n, \|\cdot\|_\infty).$$

Table 3: Descriptive statistics of benchmark datasets

|  | METABRIC | RotGBSG | FLCHAIN | SUPPORT | MIMIC-III | KKBOX |
|---|---|---|---|---|---|---|
| Size | 1904 | 2232 | 6524 | 8873 | 35953 | 2646746 |
| Censoring rate | 0.423 | 0.432 | 0.699 | 0.320 | 0.901 | 0.280 |
| Features | 9 | 7 | 8 | 14 | 26 | 15 |

Since $\Theta$ is a compact set on $\mathbb{R}$, by lemma B.8 and traditional volume argument, we have that $\widetilde{N}_{[]}(\frac{\varepsilon}{2c_3}, \Theta, \|\cdot\|_\infty) \leq N_{[]}(\frac{\varepsilon}{4c_3}, \Theta, \|\cdot\|_\infty) \lesssim \frac{1}{\varepsilon}$.

For $\widetilde{N}_{[]}(\frac{\varepsilon}{2c_3}, \mathcal{V}_n, \|\cdot\|_\infty)$, by lemma B.8, we have that $\widetilde{N}_{[]}(\frac{\varepsilon}{2c_3}, \mathcal{V}_n, \|\cdot\|_\infty) = \widetilde{N}(\frac{\varepsilon}{2c_3}, \mathcal{V}_n, \|\cdot\|_\infty)$. By [12, Lemma 2], there exists a constant $c_4 > 0$ such that $\|\widehat{\nu}_1 - \widehat{\nu}_2\|_\infty \leq c_4 \|\widehat{\nu}_1 - \widehat{\nu}_2\|_2^{s_h}$, which leads to $\widetilde{N}(\frac{\varepsilon}{2c_3}, \mathcal{V}_n, \|\cdot\|_\infty) \leq \widetilde{N}(\frac{\varepsilon^{1/s_\nu}}{(2c_3 c_4)^{1/s_\nu}}, \mathcal{V}_n, \|\cdot\|_2)$. By lemma B.7 we further have $\widetilde{N}(\frac{\varepsilon^{1/s_\nu}}{(2c_3 c_4)^{1/s_\nu}}, \mathcal{V}_n, \|\cdot\|_2) \leq N(\frac{\varepsilon^{1/s_\nu}}{2(2c_3 c_4)^{1/s_\nu}}, \mathcal{V}_n, \|\cdot\|_2)$. Let $c_\nu = \frac{1}{2(2c_3 c_4)^{1/s_\nu}}$, we have that $\widetilde{N}_{[]}(\frac{\varepsilon}{2c_3}, \mathcal{V}_n, \|\cdot\|_\infty) \leq N(c_\nu \varepsilon^{1/s_\nu}, \mathcal{V}_n, \|\cdot\|_2)$.

Thus, finally we can obtain that

$$N_{[]}(\varepsilon, \mathcal{G}_n, \|\cdot\|_\infty) \lesssim \frac{1}{\varepsilon} N(c_\nu \varepsilon^{1/s_\nu}, \mathcal{V}_n, \|\cdot\|_2).$$

$\square$

## C Experimental details

### C.1 Dataset summary

We report summaries of descriptive statistics of the 6 benchmark datasets used in section 5.2 in table 3.

### C.2 Details of synthetic experiments

Since the true model is assumed to be of PF form, we generate event time according to the following transformed regression model [18]:

$$\log H(\tilde{T}) = -m(Z) + \epsilon, \tag{15}$$

where $H(t) = \int_0^t e^{h(s)} ds$ with $h$ defined in (2). The error term $\epsilon$ is generated such that $e^\epsilon$ has cumulative hazard function $G_\theta$. The formulation (15) is the equivalent to (2) [18, 17, 45]. In our experiments, the covariates are of dimension 5, sampled independently from the uniform distribution over $[0, 1]$. We set $h(t) = t$ and hence $H(t) = e^t$. The function form of $m(Z)$ is set to be $m(Z) = \sin(\langle Z, \beta \rangle) + \langle \sin(Z), \beta \rangle$, where $\beta = (0.1, 0.2, 0.3, 0.4, 0.5)$. Then censoring time $C$ is generated according to

$$\log H(C) = -m(Z) + \epsilon_C, \tag{16}$$

which reuses covariate $Z$, and draws independently a noise vector $\epsilon_C$ such that the censoring ratio is controlled at around $40\%$. We generate three datasets with $n \in \{1000, 5000, 10000\}$ respectively.

**Hyperparameter configurations** We specify below the network architectures and optimization configurations used in all the tasks:

**PF scheme:** For both $\widehat{m}$ and $\widehat{h}$, we use 64 hidden units for $n = 1000$, 128 hidden units for $n = 5000$ and 256 hidden units for $n = 10000$. We train each model for 100 epochs with batch size 128, optimized using Adam with learning rate 0.0001, and no weight decay.

**FN scheme:** For both $\widehat{\nu}$, we use 64 hidden units for $n = 1000$, 128 hidden units for $n = 5000$ and 256 hidden units for $n = 10000$. We train each model for 100 epochs with batch size 128, optimized using Adam with learning rate 0.0001, and no weight decay.

### C.3 Details of public data experiments

**Dataset preprocessing** For METABRIC, RotGBSG, FLCHAIN, SUPPORT and KKBOX dataset, we take the version provided in the `pycox` package [46]. We standardize continuous features into zero mean and unit variance and do one-hot encodings for all categorical features. For the MIMIC-III dataset, we follow the preprocessing routines in [55] which extracts 26 features. The event of interest is defined as the mortality after admission, and the censored time is defined as the last time of being discharged from the hospital. The definition is similar to that in [64]. But since the dataset is not open sourced, according to our implementation the resulting dataset exhibits a much higher censoring rate (90.2% as compared to 61.0% as reported in the SODEN paper [64]). Since the major purpose of this paper is for the proposal of the NFM framework, We use our own version of the processed dataset to further verify the predictive performance of NFM.

**Hyperparameter configurations** We follow the general training template that uses MLP as all nonparametric function approximators (i.e., $\widehat{m}$ and $\widehat{h}$ in the PF scheme, and $\widehat{\nu}$ in the FN scheme), and train for 100 epochs across all datasets using Adam as the optimizer. The tunable parameters and their respective tuning ranges are reported as follows:

**Number of layers (network depth)** We tune the network depth $L \in \{2, 3, 4\}$. Typically, the performance of two-layer MLPs is sufficiently satisfactory.

**Number of hidden units in each layer (network width)** We tune the network width $W \in \{2^k, 5 \le k \le 10\}$.

**Optional dropout** We optionally apply dropout with probability $p \in \{0.1, 0.2, 0.3, 0.5, 0.7\}$.

**Batch size** We tune batch size within the range $\{128, 256, 512\}$, in the KKBOX dataset, we also tested with larger batch sizes $\{1024\}$.

**Learning rate and weight decay** We tune both the learning rate and weight decay coefficient of Adam within range $\{0.01, 0.001, 0.0001\}$.

**Frailty specification** We tested gamma frailty, Box-Cox transformation frailty, and IGG($\alpha$) frailty with $\alpha \in \{0, 0.25, 0.75\}$. Here note that IGG(0.5) is equivalent to gamma frailty. We also empirically tried to set $\alpha$ to be a learnable parameter and found that this additional flexibility provides little performance improvement regarding the datasets used for evaluation.

### C.4 Implementations

We use `pytorch` to implement NFM. **The source code is provided in the supplementary material**. For the baseline models:

- We use the implementations of CoxPH, GBM, and RSF from the `sksurv` package [54], for the KKBOX dataset, we use the XGBoost library [10] to implement GBM and RSF, which might yield some performance degradation.
- We use the `pycox` package to implement DeepSurv, CoxTime, and DeepHit models.
- We use the official code provided in the SODEN paper [64] to implement SODEN.
- We obtain results of SuMo and DeepEH based on our re-implementations.

## D Additional experiments

### D.1 Recovery assessment of $m(Z)$ in PF scheme

We plot empirical recovery results targeting the $m$ function in (2) in figure 2. The result demonstrates satisfactory recovery with a moderate amount of data, i.e., $n \ge 1000$.

### D.2 Recovery assessment of survival functions

To assess the recovery performance of NFM with respect to survival functions, we consider the following setup: under the same data generation framework as in section C.2, we compute the test feature $\bar{Z}$ as the sample mean of all the 100 hold-out test points. And plot $\widehat{S}(t|\bar{Z})$ against the ground

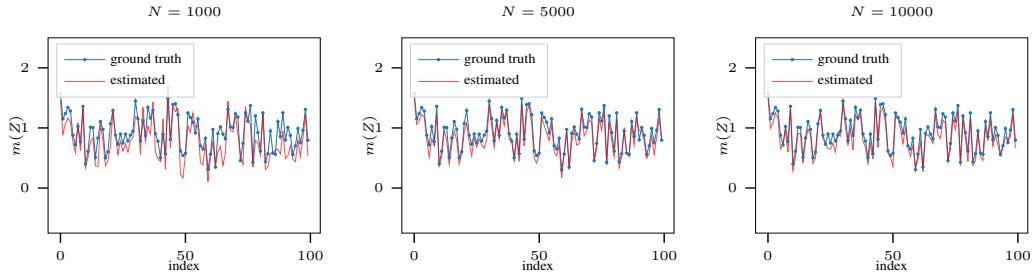

Figure 2: Visualizations of synthetic data results under the PF scheme of NFM framework, regarding empirical recovery of the $m$ function in (2)

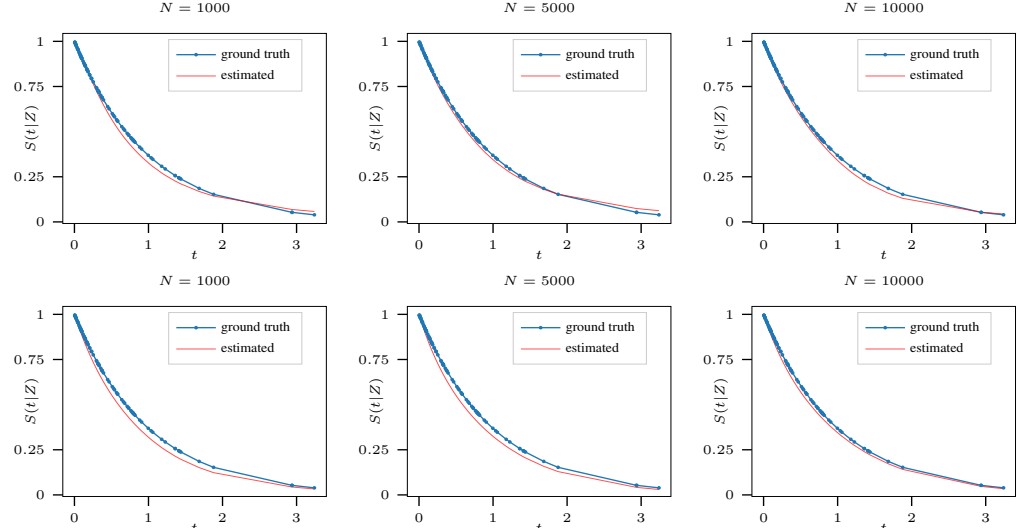

Figure 3: Visualizations of synthetic data results under the NFM framework. The plots in the first row compare the empirical estimates of the survival function $S(t|\bar{Z})$ against its true value with $\bar{Z}$ being the average of the features of the 100 hold-out points, under the PF scheme. The plots in the second row are obtained using the FN scheme, with analogous semantics to the first row.

truth $S(t|\bar{Z})$ regarding both PF and FN schemes. The results are shown in figure 3. The results suggest that both scheme provides accurate estimation of survival functions when the sample size is sufficiently large.

### D.3 Numerical results of the synthetic experiments

Following [75], we report the relative integrated mean squared error (RISE) of the estimated survival function against the ground truth and list the results in table . The reults suggest that the goodness of fit becomes better with a larger sample size. Moreover, since the true model in the simulation is generated as an PF model, we found PF to perform slightly better than FN, which is reasonable since the inductive bias of PF is more correct in this setup.

Table 4: RISE of the estimated survival function in synthetic experiments

|        | $N = 1000$ | $N = 5000$ | $N = 10000$ |
|--------|-----------|-----------|------------|
| NFM-PF | 0.0473    | 0.0145    | 0.0137     |
| NFM-FN | 0.0430    | 0.0184    | 0.0165     |

Table 5: Survival prediction results measured in C-index (%) on all the 6 benchmark datasets. In each column, the **boldfaced** score denotes the best result and the underlined score represents the second-best result. The average rank of each model is reported in the rightmost column. We did not manage to obtain reasonable results for DeepEH and SODEN on two larger datasets MIMIC-III and KKBOX, and we set corresponding ranks to be the worst on those datasets.

| Model | METABRIC | RotGBSG | FLCHAIN | SUPPORT | MIMIC-III | KKBOX | Ave. Rank |
|---|---|---|---|---|---|---|---|
| CoxPH | $63.42_{\pm1.81}$ | $66.14_{\pm1.46}$ | $79.09_{\pm1.11}$ | $56.89_{\pm0.91}$ | $74.91_{\pm0.00}$ | $83.01_{\pm0.00}$ | 11.33 |
| GBM | $64.02_{\pm1.79}$ | $67.35_{\pm1.16}$ | $\mathbf{79.47}_{\pm1.08}$ | $61.46_{\pm0.80}$ | $75.20_{\pm0.00}$ | $85.84_{\pm0.00}$ | 7.17 |
| RSF | $64.47_{\pm1.82}$ | $67.33_{\pm1.34}$ | $78.75_{\pm1.07}$ | $61.63_{\pm0.84}$ | $75.47_{\pm0.17}$ | $85.79_{\pm0.00}$ | 8.00 |
| DeepSurv | $63.95_{\pm2.12}$ | $67.20_{\pm1.22}$ | $79.04_{\pm1.14}$ | $60.91_{\pm0.85}$ | $80.08_{\pm0.44}$ | $85.59_{\pm0.08}$ | 8.50 |
| CoxTime | $66.22_{\pm1.69}$ | $67.41_{\pm1.35}$ | $78.95_{\pm1.01}$ | $61.54_{\pm0.87}$ | $78.78_{\pm0.62}$ | $\mathbf{87.31}_{\pm0.24}$ | 5.00 |
| DeepHit | $66.33_{\pm1.61}$ | $66.38_{\pm1.07}$ | $78.48_{\pm1.09}$ | $\mathbf{63.20}_{\pm0.85}$ | $79.16_{\pm0.59}$ | $86.12_{\pm0.26}$ | 6.50 |
| DeepEH | $66.59_{\pm2.00}$ | $\mathbf{67.93}_{\pm1.28}$ | $78.71_{\pm1.44}$ | $61.51_{\pm1.04}$ | – | – | 6.33 |
| SuMo-net | $64.82_{\pm1.80}$ | $67.20_{\pm1.31}$ | $79.28_{\pm1.02}$ | $62.18_{\pm0.78}$ | $76.23_{\pm1.06}$ | $84.77_{\pm0.02}$ | 7.00 |
| SODEN | $64.82_{\pm1.05}$ | $66.97_{\pm0.75}$ | $79.00_{\pm0.96}$ | $61.10_{\pm0.59}$ | – | – | 10.17 |
| SurvNode | $64.64_{\pm4.91}$ | $67.30_{\pm1.65}$ | $76.11_{\pm0.98}$ | $55.37_{\pm0.77}$ | – | – | 11.50 |
| DCM | $65.76_{\pm1.25}$ | $66.75_{\pm1.35}$ | $78.61_{\pm0.79}$ | $62.19_{\pm0.95}$ | $76.45_{\pm0.34}$ | $83.48_{\pm0.07}$ | 8.33 |
| DeSurv | $65.88_{\pm2.02}$ | $67.30_{\pm1.45}$ | $78.97_{\pm1.64}$ | $61.47_{\pm0.97}$ | $\mathbf{80.97}_{\pm0.30}$ | $86.11_{\pm0.05}$ | 5.67 |
| **NFM-PF** | $64.98_{\pm1.87}$ | $\underline{67.77}_{\pm1.35}$ | $\underline{79.45}_{\pm1.03}$ | $61.33_{\pm0.83}$ | $79.56_{\pm0.15}$ | $86.23_{\pm0.01}$ | $\underline{4.67}$ |
| **NFM-FN** | $\mathbf{66.63}_{\pm1.82}$ | $67.73_{\pm1.29}$ | $79.29_{\pm0.93}$ | $\underline{62.21}_{\pm0.41}$ | $\underline{80.18}_{\pm0.20}$ | $\underline{86.61}_{\pm0.01}$ | **2.16** |

Table 6: Relative improvement of NFM models in comparison to their non-frailty counterparts, measured in IBS, INBLL, and C-index.

| Dataset | NFM-PF vs DeepSurv | | | NFM-FN vs SuMo-net | | |
|---|---|---|---|---|---|---|
| | IBS | INBLL | C-index | IBS | INBLL | C-index |
| METABRIC | +1.33% | +1.56% | +1.61% | +2.30% | +3.08% | +2.79% |
| RotGBSG | +1.11% | +0.95% | +0.84% | +0.62% | +0.40% | +0.79% |
| FLCHAIN | +1.29% | +1.32% | +0.52% | +0.20% | +0.27% | +0.01% |
| SUPPORT | +0.31% | +0.23% | +0.69% | +2.22% | +1.76% | +0.05% |
| MIMIC-III | +12.38% | +12.15% | −0.64% | +6.18% | +5.56% | +5.18% |
| KKBOX | +2.56% | +0.51% | +0.75% | +8.20% | +10.38% | +2.17% |

### D.4 Performance evaluations under the concordance index (C-index)

The concordance index (C-index) [1] is yet another evaluation metric that is commonly used in survival analysis. The C-index estimates the probability that, for a random pair of individuals, the predicted survival times of the two individuals have the same ordering as their true survival times. Formally, C-index is defined as

$$\mathcal{C} = \mathbb{P}\left[\widehat{S}(T_i \mid Z_i) < \widehat{S}(T_j \mid Z_j) \mid T_i < T_j, \delta_i = 1\right]. \tag{17}$$

We report performance evaluations based on C-index over all the 6 benchmark datasets in table 5. From table 5, it appears that there's no clear winner regarding the C-index metric across the 6 selected datasets. We conjecture this phenomenon to be closely related to the loose correlation between the C-index and the likelihood-based learning objective, as was observed in [56]. Therefore we compute the average rank of each model as an overall assessment of performance, as illustrated in the last column in table 5. The results suggest that the two NFM models perform better on average.

### D.5 Benefits of frailty

We compute the (relative) performance gain of NFM-PF and NFM-FN, against their non-frailty counterparts, namely DeepSurv [42] and SuMo-net [56] based on results in table 1, table 2 and table 5. The results are shown in table 6 The results suggest solid improvement in incorporating frailty, especially for IBS and INBLL metrics, as the relative increase in performance could be over 10% for both NFM models. For the IBS and INBLL metrics, the performance improvement is consistent across all datasets. The only performance degradation appears on the MIMIC-III dataset evaluated under C-index. This phenomenon is also understandable: Since the DeepSurv model utilized a variant

of partial likelihood (PL) for model training, as previous works [62] pointed out that PL type objective is closely related to the ranking problem. As C-index could be considered a certain type of ranking measure, it is possible that DeepSurv obtains better ranking performance than NFM-type models which are trained using scale-sensitive likelihood objective.

## D.6 Limitations

In this section we discuss the limitations of this paper form both theoretical and empirical standpoints.

**Theoretical limitations** As we have established formal statistical guarantees regarding the estimation properties of NFM, the guanrantees do not necessarily imply that NFM perform well on prediction tasks under metrics such as IBS and INBLL. Following the spirit of classical learning theory [58], for prediction guarantees it is ideal to directly optimize the underlying metric or its surrogates, which is difficult in survival problems since the metrics involve both a model over the event time and a working model on the censoring time. So far as we have noticed, the only effort that aims to address this issue is the method of inversely-weighted survival games [33]. However, the authors in [33] did not provide rigorous learning-theoretic statements, which is a promising research direction for future works.

**Empirical limitations** While NFM is shown to be a competitive survival model for prognosis empirically, we have observed from the empirical results that we can hardly obtain *statistically significant improvements* over the baselines, which is a common problem exhibited in previous works on neural survival regressions [75, 56]. We conjecture that this phenomenon is primarily due to two facts: Firstly, there lacks open-to-public large-scale survival datasets that allows scalable evaluation of neural survival models. Secondly, for most of the current available datasets, there are no authoritative train-test splits. Consequently, most experiments are done using cross-validation on moderate scale datasets, causing the resulting variability of the modeling algorithms to be relatively large. Therefore we suggest the survival analysis community to release (in a privacy-preserving manner) more large-scale, sanitized datasets equipped with standard train-test splits, which we believe will greatly benefit the state-of-the-art for neural survival modeling.

