# OpenReview forum: "Neural Frailty Machine: Beyond proportional hazard assumption in neural survival regressions"
_NeurIPS.cc/2023/Conference — NeurIPS 2023 poster_

### Official Review · Reviewer_nR17 · 2023-06-30

**Soundness:** 2 fair
**Presentation:** 2 fair
**Contribution:** 2 fair
**Rating:** 5
**Confidence:** 4

**Summary:**

The paper proposes to parametrize frailty models with neural networks and provide results on learning rates for these models.

Experiments demonstrate the efficacy of the proposed models.

**Strengths:**

1. The theoretical analysis is interesting and cool.

**Weaknesses:**

1. The paper states that: "[56] used a neural network to approximate the conditional survival function and could be thus viewed as another trivial extension of NHR." How is this not exactly the same idea but for frailty models, you're proposing a neural parametrization very much akin to [56].


**Questions:**

1. Everything is evaluated using IBS and IBLL, non-proper scores when censoring is not independent of event-times. Can you prove that the datasets you use indeed exhibit this independence. Can you provide evaluations using the likelihood, a proper score?
2. The objective seems computationally intensive to compute, can you provide computational comparisons with other methods?

---

> ### Author Rebuttal · Authors · 2023-08-08
>
> We thank the reviewer for providing insightful comments. Below we address your specific points:
> ### Q1: On evaluation with proper score metrics
> Please kindly refer to the first part of the general response for a detailed explanation. In our opinion, while right-censored log-likelihood (CLL) is a proper metric, it is not well defined for many neural models with implicit uses of the nonparametric maximum likelihood (NPMLE) method like DeepSurv and Coxtime as well as models using pseudo likelihood like DeepEH. To provide more intuitions, think of the Breslow 's estimate for the cumulative hazard used in proportional-hazard type models (CoxPH, DeepSurv, RSF...), the estimate is a step function and therefore there are no corresponding estimates for the hazard function and hinders the evaluation of CLL. Breslow's estimate could be regarded a by-product of the NPMLE construction of partial likelihood.
>
> Now if we still want to evaluate CLL, the best one can do is to evaluate some approximated version as done in [1], with the approximation itself being not theoretically-principled. This is the reason why we did not incorporate CLL into our evaluation in the first place.
>
> For a more thorough evaluation, we provide additional experiments comparing NFM and SuMo-net, which we find is the only baseline method that allows straightforward computation of CLL metric, and NFM is shown to perform slightly better in this metric in comparison to SuMo-net. As shown in Table 1 in the second part of the general response. For completeness we paste it below:
> |               | metabric         | gbsg             | flchain          | support          | mimic-iii        | kkbox            |
> |---------------|------------------|------------------|------------------|------------------|------------------|------------------|
> | SuMo-net      | -0.256(0.052)    | 0.367(0.056)     | -1.184(0.023)    | -0.673(0.023)    | -0.052(0.003)    | 0.128(0.004)     |
> | NFM(PF)       | -0.184(0.020)    | 0.378(0.054)     | -1.169(0.022)    | -0.622(0.028)    | **0.125(0.006)** | 0.647(0.003)     |
> | NFM(PF)       | **-0.148(0.027)**| 0.369(0.046)     | -1.166(0.026)    | **-0.588(0.036)**| -0.026(0.001)    | **0.786(0.009)** |
> Table 1: Comparison of NFM with SuMo-net in the CLL metric
>
> ## Q2: On the computational complexity of using Clenshaw-Curtis quadrature
> Please kindly refer to the second part of the general response for a detailed explanation. We illustrate in Table 2 and Table 3 in the second part of the general response that using $10$ discretization steps in the quadrature method suffices for competitive performance, with the running time comparable to efficient methods like SuMo-net and being much faster than kernel approximation methods like DeepEH.  For completeness we paste the results below:
>
> |dataset |          |NFM(PF)   |          |          |NFM(FN)   |          |SuMo-net|DeepEH  |
> |--------|----------|----------|----------|----------|----------|----------|--------|--------|
> |        |steps = 10|steps = 20|steps = 50|steps = 10|steps = 20|steps = 50|        |        |
> |metabric|9.7s      | 12.5s    |17.7s     |11.4s     | 19.4s    | 36.1s    | 13.58s | 85.6s  |
> Table 2: Running time comparisons of using different number of discretization steps, along with two baselines
>
> |          |           | NFM(PF)   |           |           | NFM(FN)   |           |
> |----------|-----------|-----------|-----------|-----------|-----------|-----------|
> |          |cindex     |ibs        |inbll      |cindex     |ibs        |inbll      |
> |steps = 10|65.16(1.46)|16.28(0.76)|49.02(2.29)|66.82(1.62)|16.03(0.87)|47.96(2.53)|
> |steps = 20|65.16(1.46)|16.28(0.76)|49.02(2.29)|66.63(1.68)|16.07(0.84)|48.04(2.44)|
> |steps = 50|65.13(1.46)|16.28(0.76)|49.03(2.29)|66.79(1.49)|16.10(0.83)|48.10(2.43)|
> Table 3: Performance comparisions of using different number of discretization steps
>
> 1] Rindt, David, et al. "Survival regression with proper scoring rules and monotonic neural networks." International Conference on Artificial Intelligence and Statistics. PMLR, 2022.

---

> > ### Comment · Reviewer_nR17 · 2023-08-16
> > **Re: Rebuttal**
> >
> > Thanks for the response, raising to 5.

---

### Official Review · Reviewer_82in · 2023-07-04

**Soundness:** 3 good
**Presentation:** 3 good
**Contribution:** 3 good
**Rating:** 7
**Confidence:** 3

**Summary:**

The authors propose a neural architecture to estimate the survival function for observations of survival times and censored survival times. The authors describe a methodology to include heterogeneity among the population by including the frailty component. The authors then theoretically and empirically illustrate properties of their approach. The theoretical results are used to provide additional justification for why a neural network architecture is appropriate for this kind of problem. Empirical results are added to place into context the performance of the proposed methods with respect to existing methods.


**Strengths:**

The paper provides a thorough analysis on using neural networks for survival estimation beyond the analyses present in previous works. The authors also consider frailty which they naturally include within their modelling framework, removing some of the homogeneity assumptions of previous works. The empirical results suggest that the method performs well in the considered scenarios. I did not check the details of the proofs, but the theoretical component seems useful and original for predicting convergence rates of the estimator and applicable to other similar estimators. Finally, the writing is clear and easy to follow.

**Weaknesses:**

The main weakness involves the empirical evaluation of the paper. None of the empirical gains are major when considering the empirical experiments. This is expected with real world data since it's difficult to estimate the counterfactual (i.e. maybe the survival time could have been much longer/shorter). I would suggest to add an empirical result that compares the methods in idealized settings where the true survival distribution is known and that the metrics can be appropriately compared.

Another potential weakness involves the theoretical guarantees. If I'm understanding correctly, the theory applies in the case where the networks are well-learned, which is still subject to the issues associated with neural network optimization. It may be good to emphasize this in the text.

**Questions:**

How does the empirical convergence compare with the theoretical?

The authors mention using Clenshaw-Curtis as the integrator, how does the parameters of the integrator (e.g. step size) affect the performance of the method? Is there some numerical bottleneck related to this integration?

In the theoretical results, the MLPs scale with the number of samples, is there a rigorous connection to the number of parameters in the network and the number of samples? Additionally, was this enforced in the experimental setup, I could not tell if this was the case in the hyper parameter section of the appendix?

Does the method extend to representing the joint distribution of multiple survival times? E.g. something like $P(T_1 > t_1, \ldots, T_d > t_d \mid Z)$?

Was a nonparametric model for the frailty component examined, i.e. could $f_\theta$ be represented by some generative model constrained to be positive?

**Limitations:**

The authors did mention limitations in the appendix. Alongside their notes, it may be good to get a review of where the authors think that their method is appropriate and where other methods would be more appropriate. Additionally, it would be nice to see conditions where the method fails to get a better idea of the properties of the method. This could possibly come from some of the theoretical assumptions being violated (e.g. it’s unclear which ones are really necessary in practice versus for proving the theorems).

---

> ### Author Rebuttal · Authors · 2023-08-08
>
> We thank the reviewer for providing insightful comments. Below we address your specific points:
> ### Q1: Evaluation of idealized settings
> According to our understanding, if the true surival time is always observed (i.e., no censoring effects), then the problem boils down to an ordinary regression problem and there are many appropriate metrics and learning objectives that allow a formal study of generalization. In our synthetic experiments, the true survival is known and we provided a pictorial illustration in figure 3 at appendix D that assesses the recovery of the true surival function. We also computed the relative integrated mean squared error (RISE) corresponding to different sample sizes. The results are listed in the following table, showing that the goodness of fit becomes better with a larger sample size.
> |       |N=1000|N=5000|N=10000|
> |-------|------|------|-------|
> |NFM(PF)|0.0473|0.0145|0.0137 |
> |NFM(FN)|0.0430|0.0184|0.0165 |
> Table 1. RISE for the synthetic experiments
>
> ### Q2: Theoretical guarantees and optimization issues
> This is a very good point. Throughout this paper, we rely on the empirical observation that state-of-the-art stochastic optimization algorithms like Adam produce satisfactory results for neural models, and set aside the convergence issue (in the language of optimization). A careful analysis of the optimization landscape of the NFM objective is beyond the scope of the paper, but is a valuable topic and worth future explorations. We will add a discussion in the camera-ready version of the paper.
> Besides, regarding your comments "..networks are well-learned..", according to our understanding, to place ourselves in a machine learning context the issue is somewhat related to some notion of *generalization* in survival analysis, which is to our knowledge an open problem. In some recently proposed frameworks like [6] it might offer opportunities to study formal generalization in survival analysis, which is a promising future direction.
>
> ### Q3: Comparison of empirical and theoretical convergence
> In this paper, we establish statistical guarantees using the Hellinger distance which is hard to empirically evaluate [1]. Therefore we instead compute a intuitive metric RISE and reported in Table 1. We can see from the results that the goodness of fit becomes better with a larger sample size.
>
> ### Q4: On the complexity of using Clenshaw-Curtis quadrature
> Please kindly refer to the second part of the general response for a detailed explanation. We illustrate in Table 2 and Table 3 in the second part of the general response that using $10$ discretization steps in the quadrature method suffices for competitive performance, with the running time comparable to efficient methods like SuMo-net and being much faster than kernel approximation methods like DeepEH.
>
> ### Q5: On the scale of the network versus sample size
> In our theoretical analysis, we construct sieve spaces as a set of MLPs with depth and number of parameters that grow with the sample size at certain speed. This is actually the definition of sieve method [2] that construct a series of parameter spaces that eventually becomes dense in the target function space (the Holder ball defined in (8) in the paper). The construction is mostly driven by approximation-theoretic type arguments [3] and serves as a guide instead of enforcement for empirical hyperparameter choice. As you have pointed out, there are multiple additional factors that may affect the empirical results including optimization issues and training strategy. In our experiments, the final models are selected using early stopping on validation datasets.
>
> ### Q6: On extension to multiple survival times
> This is another very interesting question. Actually the frailty model is even more appropriate for scenarios where multiple surivival times are present, and we may use the powerful tool of frailty to describe the correlation structure among the individual survival times. Such extension requires carefully modeling the dependence structure [5] and will be left to further studies. See also remark 3.1 in the paper.
>
> ### Q7: Is it possible for a nonparametric model of frailty?
> This is yet another interesting and intriguing question. In general, if we allow the frailty transform to be nonparametric, it becomes statistically hard to separate the effect of the frailty transform and its input argument, as they are both infinite-dimensional and extremely expressive (see definition (3) in the paper). Therefore to study such kind of extensions we may need some extra restrictions on the generating process [5]. That said, we did try this option during our empirical evaluations and find this to perform pretty well empirically (comparable but not significantly surpassing parametric NFM). However as they are very subtle in theory, we chose not to report this line of results.
>
> [1] Sreekumar, Sreejith, and Ziv Goldfeld. "Neural estimation of statistical divergences." The Journal of Machine Learning Research 23.1 (2022): 5460-5534.
> [2] Chen, Xiaohong. "Large sample sieve estimation of semi-nonparametric models." Handbook of econometrics 6 (2007): 5549-5632.
> [3] Yarotsky, Dmitry. "Error bounds for approximations with deep ReLU networks." Neural Networks 94 (2017): 103-114.
> [4] Parner, Erik. "Asymptotic theory for the correlated gamma-frailty model." The Annals of Statistics 26.1 (1998): 183-214.
> [5] Tang, Weijing, et al. "Survival analysis via ordinary differential equations." Journal of the American Statistical Association (2022): 1-16.
> [6] Han, Xintian, et al. "Inverse-weighted survival games." Advances in neural information processing systems 34 (2021): 2160-2172.

---

> > ### Comment · Reviewer_82in · 2023-08-18
> >
> > I thank the authors for their response. After seeing the other reviews and the response, I would like to keep my score. However, it would be good to make a statement regarding how the proposed method is more than just a straightforward extension of existing methods (e.g. highlighting the major contributions that make the frailty model more challenging than a simple extension) as well as including additional numerical results demonstrating when the method should perform well. Of course, only so much is possible during the response period, so it would be good to augment the final version of the paper with some of these additional experiments.

---

### Official Review · Reviewer_PAZm · 2023-07-06

**Soundness:** 3 good
**Presentation:** 3 good
**Contribution:** 2 fair
**Rating:** 4
**Confidence:** 3

**Summary:**

The frailty model is one of the popular models in survival analysis, and it is an extension of the classical Cox proportional hazard model.   This paper extends the frailty model by using neural networks, and this paper provides the theoretical analysis of the proposed models.

**Strengths:**

+ This paper proposes two new models NFM-PF and NFM-PM by combining the frailty model and neural networks.
+ This paper provides the theoretical analysis of the proposed models to show its correctness.

**Weaknesses:**

I think that the (original) frailty model is interesting, simply because it uses slightly weaker assumptions than the classical Cox model.  Even though their difference is small, the frailty model can be significantly better than the Cox model in practical prediction performance.  This is because the Cox model is essentially a linear model and therefore a slight change (i.e., using frailty) can be a huge differentiator.  However, the proposed extensions of frailty models by using neural networks are not so interesting. We already know many neural network models for survival analysis such as DeepSurv [42] and DeepEH [75], which are extensions of the Cox and AFT models.  All of the neural network models for survival analysis are flexible enough, and the advantage of using the concept of frailty in neural network models should be minimal.

The experiments are not good enough with respect to showing the practical advantage of using frailty in neural network models.  Although the proposed models were compared with many existing models, the performance differences seem coming from the differences of the neural network architectures (e.g., the network structures, the number of layers, the number of neurons in each layer, and many other implementation details) rather than the frailty.  Since the goal of the experiments is to show the advantage of using frailty in the neural network models, the authors should compare the two neural network models with and without frailty by using (almost) the same neural network architecture.

Moreover, the authors should have used the right-censored log-likelihood, which is shown to be a strictly proper scoring rule in [56], as the evaluation metric instead of IBS and IBNLL in the experiments.  The statement “Both IBS and INBLL are proper scoring rules” in Line 301 is not true, and it is shown that both IBS and INBLL are not proper in [56].

**Questions:**

I have no question.

**Limitations:**

I couldn’t find any description on limitations.

---

> ### Author Rebuttal · Authors · 2023-08-08
>
> We thank the reviewer for providing insightful comments. Below we address your specific points:
> ## Q1: The motivation of frailty and its advantage
> Firstly, we would like to emphasize that random effect (called frailty), which serves as a principled tool to model *unobserved heterogeneity*, has played an important role in modern survival analysis which is beyond a simple relaxation of proportional hazard assumption. There has been significant and active research concerning the addition of random effect to survival models. The random effect can describe risk or frailty for distinct categories, such as individuals or famlies. There eixst literature on providing theoretical and practical motivation for frailty models by discussing the impact of heterogeneity on analyses. In the influential article [1], Aalen showed that with a sufficiently strong frailty the population relative risk can go from r to 1/r over time, even though the true relative risk stays at r.  As you have pointed out, DeepSurv and DeepEH are flexible models compared to CoxPH. However, DeepSurv still encodes the belief of proportional hazard, and frailty models relax this assumption and is strictly more powerful than DeepSurv.
>
> Secondly, while the algorithmic extension of incorporating frailty is straightforward, **it is highly nontrivial to establish statistical guarantees**. In this paper the goal is to introduce a rigorous statistical model with provable guarantees, which was lacked in many previous works. As was pointed out in the paper, the current theoretical developments on neural survival models rely on the fact that the loss function is well-controlled by the L2 loss [2, 3], which is not directly applicable to our model due to the flexibility in choosing the frailty transform. **In this paper, we developed a framework that further extends the proof strategy of previous works to more general losses, which we think is a significant contribution to the theoretical developments of neural survival models**.
>
> Finally, regarding your comments " the authors should compare the two neural network models with and without frailty by using (almost) the same neural network architecture.", we have provided such comparison in Table 5 in appendix D. The result shows the benefits of frailty in terms of predictive performance. Next we provide a further study on the presence of frailty:
> While the precise notion of explaining frailty seems to go beyond the scope of our paper, we conducted heuristic experiments to investigate the **strength of frailty effects** in the real-world data used in our paper. The methodology is to use the bootstrap test [4]. In particular, we construct $R=10$ bootstrap samples of each dataset, and then computes the frailty parameter estimates of each bootstrap replicate and report the means and standard deviations of bootstrap estimates. The test allows standard asymptotics if the estimate itself is asymptotically normal which is to be established in our future study. We treat the method as being heuristic. The results are summarized in the following table:
> |               |metabric |gbsg |flchain|support|mimic-iii|kkbox    |
> |---------------|---------|-----|-------|-------|---------|---------|
> |bootstrap mean |0.650    |0.569|0.678  |1.945  |0.857    |1.391    |
> |bootstrap stdev|0.017    |0.017|0.070  |0.118  |0.023    |0.066    |
> |mean/stdev     |**38.23**|33.47|9.69   |16.48  |**37.26**|21.08    |
> Table 1: bootstrap tests against the presence of frailty
>
> Here we use (bootstrap mean)/(bootstrap stdev) as a intuitive measure of the presence of frailty effect (the rationale is similar to the Z-score in standard hypothesis testing). According to the table, the effect of frailty is mostly evident in metabric and mimic-iii dataset, which is coherent with the gain of frailty reported in Table 5 in appendix D of the paper.
>
> While there are many off-the-shelf neural survival models that are flexible and performative, only very few of them come with formal statistical guarantees (We believe some of them actually are theoretically sound without proofs). And the existing proof strategy has its own limitations. In this paper we make contributions to the theoretical developments via adopting new proof strategies to establish statistical guarantees, and we believe the strategy could be used to derive results for other approaches.
>
> ## Q2: On evaluation with proper score metrics
> Please kindly refer to the first part of the general response for a detailed explanation. First we would like to clarify that the statement in the paper was "Both IBS and INBLL are proper scoring rules if the censoring times and survival times are independent". In our opinion, while right-censored log-likelihood (CLL) is a proper metric, it is not well defined for many neural models with implicitly uses the method of nonparametric maximum likelihood (NPMLE) like DeepSurv and Coxtime, and the best thing can do is to evaluate some approximated version, with the approximation itself being not theoretically-principled. For a more thorough evaluation, we provide additional experiments comparing NFM and SuMo-net, which we find is the only baseline method that allows straightforward computation of CLL metric, and NFM is shown to perform slightly better in this metric in comparison to SuMo-net. As shown in Table 1 in the first part of the global response.
>
> [1] Aalen, Odd O. "Heterogeneity in survival analysis." Statistics in medicine 7.11 (1988): 1121-1137.
> [2]. Zhong, Qixian, Jonas W. Mueller, and Jane-Ling Wang. "Deep extended hazard models for survival analysis." Advances in Neural Information Processing Systems 34 (2021): 15111-15124.
> [3]. Zhong, Qixian, Jonas Mueller, and Jane-Ling Wang. "Deep learning for the partially linear Cox model." The Annals of Statistics 50.3 (2022): 1348-1375.
> [4]. Davison, Anthony Christopher, and David Victor Hinkley. Bootstrap methods and their application. No. 1. Cambridge university press, 1997.

---

> > ### Comment · Reviewer_PAZm · 2023-08-19
> >
> > Thank you for the authors' comments.  I acknowledge that I have read all of the four reviews and the authors' comments.  Having read the comments, I keep my score (i.e., I'm on borderline and I'm inclined to rejection), because I'm not convinced why the proposed methods *significantly* outperform the prior methods in the experiments even though the technical difference between the proposed methods and the prior methods is so small (i.e., using frailty).
> >
> > I would suggest clarifying all the hyper parameters (both for the proposed methods and the prior methods) and other details (e.g., python version) used in your experiments so that researchers can reproduce your experimental results.  Note: I'm *not* requesting additional comments/experiments in this rebuttal phase, but I would suggest adding the information in the camera-ready version (if this paper is accepted) or the future manuscript (if this paper is rejected).

---

> > > ### Author Response · Authors · 2023-08-19
> > > **Some further clarifications**
> > >
> > > Thank you for the response. We will incorporate your suggestions in revised versions of the paper. Additionally, we provided experiment details in appendix C.3 of the paper regarding the tuning of hyper-parameters.
> > >
> > > Moreover, we would like to point out that **we did not make the conclusion that NFM is significantly better than previous SOTA, but mostly comparable and sometimes slightly surpassing** regarding the benchmark datasets which we used. So far as we have noticed, the empirical results on the chosen datasets rarely exhibit large performance gaps: Even compared to the vanilla CoxPH, the current SOTA methods (considering the best performing one) only shows statistically significant advantage (at level 0.05) on 4 out of 6 datasets.
> > >
> > > Finally, frailty is a flexible framework has many possible parameterizations which we elaborate in appendix A of the paper. Could you tell us a little bit more on why you insist that the difference between frailty formulations and previous approaches is "so small"?

---

### Official Review · Reviewer_U6h7 · 2023-07-10

**Soundness:** 3 good
**Presentation:** 3 good
**Contribution:** 2 fair
**Rating:** 4
**Confidence:** 3

**Summary:**

The authors propose a framework for survival regression called neural frailty machine. They have shown that most of the existing methods are a special case of the proposed framework. Also, they have drawn statistical convergence guarantees for the proposed model. The experiments show marginal improvement compared to existing models in terms of some metrics.

**Strengths:**

The paper is well-written and has clear motivation. The overall approach is intuitive and makes sense (although I haven't checked the proofs). The main contributions are introducing frailty variable and the the statistical guarantees drawn for the proposed approaches.

**Weaknesses:**

The main weakness is lack of discussion/explanation around the results and the two proposed approaches (PF and FN). More on this in the next section.

**Questions:**

- What is the intuition behind introducing PF? It is mentioned (in third page, line 127) that "Proportional-style assumption over hazard functions has been shown to be a useful inductive bias in survival analysis". If this type of function approximation has been already shown deployed in survival analysis, then it wouldn't be very novel.
- Checking results, sometimes FN performs better, while PF works better other times. Why? Are there any explanations for each dataset?
- Is it possible to report numerical results for Figure 1, so we can compare PF and FN?
 - Addition of frailty variable (w) has marginal effect (<~1%) on most of the dataset (Table 5 in Appendix) while having huge effect on one the datasets. Any explanations for this discrepancy?

Overall, I think more discussion around numerical results is needed.

**Limitations:**

Authors discussed some the limitations in the appendix.

---

> ### Author Rebuttal · Authors · 2023-08-08
>
> We thank the reviewer for providing insightful comments. Below we address your specific points
> ## Q1: The intuition of proportional frailty model
> Firstly, we would like to emphasize that random effect (called frailty), which serves as a principled tool to model *unobserved heterogeneity*, has played an important role in modern survival analysis. There has been significant and active research concerning the addition of random effect to survival models. The random effect can describe risk or frailty for distinct categories, such as individuals or famlies. There eixst literature on providing theoretical and practical motivation for frailty models by discussing the impact of heterogeneity on analyses. Below is a quote from [1]:
> > It is a basic observation of medical statistics that individuals are dissimilar ... Still, there is tendency to regard this variation as a nuisance, and not as something to be considered seriously in its own right. Statisticians are often accused of being more interested in averages, and there is some truth to this.
>
> In the influential article [1], Aalen showed that with frailty, the population relative risk can go from r to 1/r over time, even though the true relative risk stays at r. Frailty is shown to be very useful to adjust for heterogeneity in real data.
>
> Secondly, while the algorithmic extension of incorporating frailty is straightforward, **it is highly nontrivial to establish statistical guarantees**. In this paper the goal is to introduce a rigorous statistical model with provable guarantees, which was lacked in many previous works. As was pointed out in the paper, the current theoretical developments on neural survival models rely on the fact that the loss function is well-controlled by the L2 loss [2, 3], which is not directly applicable to our model due to the flexibility in choosing the frailty transform. **In this paper, we developed a framework that further extends the proof strategy of previous works to more general losses, which we think is a significant contribution to the theoretical developments of neural survival models**.
>
> ## Q2: The benefits of frailty and the implications from empirical evaluations
> Here we answer the second and fourth question simultaneously. From an approximation point of view, FN is a more general scheme than PF. However it has been shown in recent developments of deep learning that the universality of some model does not necessarily imply its empirical effectiveness, but instead inductive bias matters, which is sometimes hard to formally quantify. At this stage we would like to argue that the additional inductive bias provided by the PF scheme may sometimes show its usefulness on specific datasets.
> While the precise notion of explaining frailty seems to go beyond the scope of our paper, we conducted heuristic experiments to investigate the **strength of frailty effects** in the real-world data used in our paper. The methodology is to use the bootstrap test [4]. In particular, we construct $R=10$ bootstrap samples of each dataset, and then computes the frailty parameter estimates of each bootstrap replicate and report the means and standard deviations of bootstrap estimates. The test allows standard asymptotics if the estimate itself is asymptotically normal which is to be established in our future study. Therefore we treat the method as being heuristic. The results are summarized in the following table:
> |               |metabric |gbsg |flchain|support|mimic-iii|kkbox    |
> |---------------|---------|-----|-------|-------|---------|---------|
> |bootstrap mean |0.650    |0.569|0.678  |1.945  |0.857    |1.391    |
> |bootstrap stdev|0.017    |0.017|0.070  |0.118  |0.023    |0.066    |
> |mean/stdev     |**38.23**|33.47|9.69   |16.48  |**37.26**|21.08    |
> Table 1: bootstrap tests against the presence of frailty
>
> Here we use (bootstrap mean)/(bootstrap stdev) as an intuitive measure of the presence of frailty effect (the rationale is similar to the Z-score in standard hypothesis testing). According to the table, the effect of frailty is mostly evident in metabric and mimic-iii dataset, which is coherent with the gain of frailty reported in Table 5 in appendix D of the paper.
>
> ## Q3: Numerical evaluations of the synthetic experiments
> Following [5], we report the relative integrated mean squared error (RISE) of the estimated survival function against the ground truth and list the results in the following table. The reults suggest that the goodness of fit becomes better with a larger sample size. Moreover, since the true model in the simulation is generated as an PF model, we found PF to perform slightly better than FN, which is reasonable since the inductive bias of PF is more correct in this setup.
> |       |N=1000|N=5000|N=10000|
> |-------|------|------|-------|
> |NFM(PF)|0.0473|0.0145|0.0137 |
> |NFM(FN)|0.0430|0.0184|0.0165 |
>
> [1] Aalen, Odd O. "Heterogeneity in survival analysis." Statistics in medicine 7.11 (1988): 1121-1137.
> [2]. Zhong, Qixian, Jonas W. Mueller, and Jane-Ling Wang. "Deep extended hazard models for survival analysis." Advances in Neural Information Processing Systems 34 (2021): 15111-15124.
> [3]. Zhong, Qixian, Jonas Mueller, and Jane-Ling Wang. "Deep learning for the partially linear Cox model." The Annals of Statistics 50.3 (2022): 1348-1375.
> [4]. Davison, Anthony Christopher, and David Victor Hinkley. Bootstrap methods and their application. No. 1. Cambridge university press, 1997.
> [5]. Zhong, Qixian, Jonas W. Mueller, and Jane-Ling Wang. "Deep extended hazard models for survival analysis." Advances in Neural Information Processing Systems 34 (2021): 15111-15124.

---

> > ### Comment · Reviewer_U6h7 · 2023-08-20
> >
> > Thanks authors for the response. Regarding Q2 results, it doesn't seem that the proposed score shows the level frailty (maybe only presence of frailty?). Because, It is not aligned with the table 5 in appendix. MIMIC-III shows significantly higher improvement while having a comparable level of frailty estimate to metabric (or gbsg).

---

> > > ### Author Response · Authors · 2023-08-21
> > > **Some further clarifications**
> > >
> > > Thanks for the response. Perhaps there are some misunderstandings and misinterpretations regarding the results in table 1 in the rebuttal thread. We would like to make a few clarifications:
> > > - Speaking in a statistical context, the meaning in the frailty parameter is usually related to the variance of the random effect. In the additional experiments related to those results in table 1, we are primarily interested in assessing the presence of frailty (as you have also pointed out in the response), which is mostly reflected through the surrogate Z-score (third row of table 1).  According to table 1, the presence of frailty is mostly significant in metabric and mimic-iii, which is coherent with table 5 in the appendix.  Here we would like to re-emphasize that this is merely empirical observations.
> > > - As far as we can tell, the mean value of the frailty parameter has no explicit connection with how NFM performs regarding the predictive metrics like Cindex, IBS and INBLL. It might be somewhat misleading of using the term **strength of frailty effects**, which we want to correct here.

---

### Author Rebuttal · Authors · 2023-08-08

We'd like to thank all the reviewers for providing insightful comments, we will integrate some of the suggestions into the camera-ready version.
We have found that there are several common issues raised by different reviewers:
- The motivation of frailty and its advantage.
- IBS/IBNLL/Cindex are not proper scoring rules in general.
- The complexity of using Clenshaw-Curtis quadrature to compute the learning objective.

Due to limited space, the first issue is postponed to the response to specific reviewers. Below we provide detailed explanations for the rest issues:

## IBS/IBNLL/Cindex are not proper scoring rules in general.
It was pointed out by several reviewers that IBS and IBNLL are improper scoring rules when the censoring time is not marginally independent with the event time. In [1] the authors advocate using censored log-likelihood (CLL) which is a proper metric. We explain below why CLL is not adopted in our paper, and provide empirical evaluations regarding this metric:
- **CLL might be not well-defined for a lot of survival models**: Many survival models are semiparametric with a nuisance parameter being assumed to lie in an infinite-dimensional functional space. The construction of the renowned partitial likelihood for the CoxPH model is a representative case of nonparametric maximum likelihood (NPMLE) (see [4] for the construction). It is well known in statistical literature that **without proper restriction, nonparametric likelihood can be unbounded** [3, 5]. In the construction of partial likelihood, the estimate of the cumulative hazard function is restricted to be a step function and there are no proper estimate of hazard function under NPMLE, thereby hindering the evaluation of CLL. However, most current survival baselines more or less uses the rationale of partial likelihood, including RSF, DeepSurv, Coxtime and many others. Besides, it is also tricky to apply CLL to models using pseudo likelihood like DeepEH. In [1], the authors used a finite-difference approximation to compute an approximate version of CLL that was not well-defined for models like DeepSurv or Coxtime. Therefore, we think that despite its properness, the applicability of CLL is limited for a fair comparison.
- **A comparison in CLL with SuMo-net**: Although CLL is not a generally applicable metric for survival models as argued above, it is applicable to the NFM framework. Therefore, we provide an additional evaluation of CLL with a comparison to SuMo-net [1], the reults are summarized in table 1 in the attachment pdf. The results suggest that NFM performs on par with SuMo-net, with statistically significantly better results (under $0.05$ significance level) on 4-out-of-6 datasets.

## The complexity of using Clenshaw-Curtis quadrature to compute the learning objective.
The only hyperparameter involved in the computation of Clenshaw-Curtis quadrature (CCQ) is the number of discretization steps which are common to many numerical integration methods. Here we provide assessments on this hyperparameter on both computational efficiency (measured in wall clock running time) and the trade-off with model performance via comparing the model performance using different number of discretization steps. All experiments are made on the metabric dataset under the optimally tuned architecture, trained for $50$ epochs using a M1 Max MacBook Pro 2021 in its cpu version. We first report the efficiency evaluation in table 2 from the attachment pdf, where we assess three configurations with $\text{steps} \in \{10, 20, 50 \}$ and compare with SuMo-net as well as DeepEH (with a computationally identical configuration of neural function approximator). From the results we find that
- PF is faster than FN and is less affected by increasing the number of integration steps. This stems from the fact that PF decouples the computation of $h$ and $m$, and we only need to evaluate the numerical integral of a one-dimensional function. For the FN scheme, integral of a multi-dimensional function is involved and is thus more time-consuming.
- With a proper choice of steps (i.e., less than $20$), both NFM schemes are comparable in efficiency to SuMo-net, which is regarded as an efficient implementation that involves only a single feedforward neural network call. Besides, NFM are much more efficient then DeepEH, since the computation of DeepEH's objective involves kernel approxiamtions and is quadratic in sample size.
Overall, **the computational cost of both NFM schemes are controllable for a moderate number of steps**. Next we study the effects on model performance. The experiments are conducted on the metabric dataset and summarized in table 3 from the attachment pdf.  We conclude from the table that even using $10$ integration steps suffices for competitive performance, and using a larger number of integration steps seems to have little extra gain. Therefore, we conclude that the additional complexity of using CCQ for evaluation of learning objective is totally affordable, especially when GPUs are deviced for parallel evaluation, the computation time might be further reduced.


[1] Rindt, David, et al. "Survival regression with proper scoring rules and monotonic neural networks." International Conference on Artificial Intelligence and Statistics. PMLR, 2022.
[2] Bickel, Peter J., et al. Efficient and adaptive estimation for semiparametric models. Vol. 4. Baltimore: Johns Hopkins University Press, 1993.
[3] Kiefer, Jack, and Jacob Wolfowitz. "Consistency of the maximum likelihood estimator in the presence of infinitely many incidental parameters." The Annals of Mathematical Statistics (1956): 887-906.
[4] Murphy, Susan A., and Aad W. Van der Vaart. "On profile likelihood." Journal of the American Statistical Association 95.450 (2000): 449-465.
[5] Zeng, Donglin, and D. Y. Lin. "Efficient estimation for the accelerated failure time model." Journal of the American Statistical Association 102.480 (2007): 1387-1396.

---

### Comment · Area_Chair_SB7u · 2023-08-15
**please engage in discussion**

Hello reviewers.

Please read the authors' response to your reviews *and to the other reviewers' reviews*, and discuss. The authors have provided reasonably detailed rebuttals so it would be great if you engage in discussion as soon as possible, especially to indicate if your opinion of the paper has changed or if you would like any sort of additional comments or clarifications.

Thanks,
Area chair

---

### Decision · Program_Chairs · 2023-09-21

**Decision:**

Accept (poster)

**Comment:**

The authors provided fairly detailed responses to the reviewers that I think sufficiently address reviewer concerns. Note that of the four reviewers, two leaned toward acceptance. Among the two reviewers who were negative (both voting for "borderline reject"), the main concerns appear to be on experimental results roughly in terms of two aspects: (i) how to know when we would benefit from modeling frailty, and (ii) how the proposed methods compare to non-frailty-based methods. The authors have in their rebuttal as well as their discussions with reviewers addressed these aspects to some extent, where I realize that fully addressing them can be quite challenging (especially for the real datasets considered, I agree with what the authors said in that even for SOTA methods, the performance gaps are often not large). The experimental results are informative even though the proposed methods do not always work the best (they work reasonably well though). That the experimental results (including the newly added ones during the author-reviewer discussion period) are not amazing does not detract from the main strength of this work which is on the theoretical development of modeling frailty with neural nets (including statistical guarantees), which I think is a valuable contribution.